# Systematic morphological and morphometric analysis of identified olfactory receptor neurons in *Drosophila melanogaster*

Cesar Nava Gonzales[1†], Quintyn McKaughan[1†], Eric A Bushong[2], Kalyani Cauwenberghs[1], Renny Ng[1], Matthew Madany[2], Mark H Ellisman[2], Chih-Ying Su[1*]

[1]Neurobiology Section, Division of Biological Sciences, University of California, San Diego, San Diego, United States; [2]National Center for Microscopy and Imaging Research, Center for Research in Biological Systems, University of California, San Diego, San Diego, United States

**Abstract** The biophysical properties of sensory neurons are influenced by their morphometric and morphological features, whose precise measurements require high-quality volume electron microscopy (EM). However, systematic surveys of nanoscale characteristics for identified neurons are scarce. Here, we characterize the morphology of *Drosophila* olfactory receptor neurons (ORNs) across the majority of genetically identified sensory hairs. By analyzing serial block-face electron microscopy images of cryofixed antennal tissues, we compile an extensive morphometric data set based on 122 reconstructed 3D models for 33 of the 40 identified antennal ORN types. Additionally, we observe multiple novel features—including extracellular vacuoles within sensillum lumen, intricate dendritic branching, mitochondria enrichment in select ORNs, novel sensillum types, and empty sensilla containing no neurons—which raise new questions pertinent to cell biology and sensory neurobiology. Our systematic survey is critical for future investigations into how the size and shape of sensory neurons influence their responses, sensitivity, and circuit function.

**\*For correspondence:**
c8su@ucsd.edu

[†]These authors contributed equally to this work

**Competing interests:** The authors declare that no competing interests exist.

## Introduction

Most insect species, including agricultural pests and disease vectors, rely on olfaction to guide critical behaviors including foraging, feeding, mating, egg-laying, and aggression (*Kim et al., 2017*). Odor detection begins with the olfactory organs, the antenna and maxillary palp, which are covered with sensory hairs, or sensilla, that encapsulate the dendrites of olfactory receptor neurons (ORNs). In *Drosophila melanogaster*, olfactory sensilla fall into four morphological classes—basiconic, coeloconic, intermediate, and trichoid—which are distinguished by their length, size, shape, and cuticular pore arrangement (*Shanbhag et al., 1999*). Notably, ORNs housed in different sensillum classes have distinct dendritic branching patterns and odor tuning properties. The outer dendritic segments of basiconic and intermediate ORNs display multiple branches, whereas the outer dendrites of coeloconic and trichoid neurons are typically unbranched (*Shanbhag et al., 1999*). While basiconic ORNs respond to a wide range of odorants, trichoid neurons respond predominantly to phero-mones, and most coeloconic neurons are selectively tuned to amines and carboxylic acids (*Dweck et al., 2015*; *Hallem and Carlson, 2006*; *Lin et al., 2016*; *Silbering et al., 2011*; *van der Goes van Naters and Carlson, 2007*; *Yao et al., 2005*). The link between sensillum class, dendritic branching pattern, and odor tuning property suggests that the morphological features of sensilla

and ORNs likely influence olfactory function. However, testing this possibility requires a systematic survey of these morphological features at nanoscales for neurons whose receptors have been identified and whose odor response profiles have been characterized.

The majority of antennal sensilla house multiple ORNs. Each neuron is assigned a designation indicating the host sensillum and its relative extracellular spike amplitude among grouped neurons. Typically two and up to four ORNs are compartmentalized in a sensillum in stereotyped combinations—whereby a neuron expressing a particular receptor is always paired with a neighboring neuron that expresses another specific receptor (*Benton et al., 2009*; *Couto et al., 2005*; *Fishilevich and Vosshall, 2005*; *Hallem and Carlson, 2006*; *Hallem et al., 2004*). For example, the Or59b receptor is expressed in the ab2A neuron, which is located in the antennal basiconic sensillum of type 2, while the 'A' indicates that its spike amplitude is greater than that of the neighboring ab2B neuron, which invariably expresses the Or85a receptor (*Hallem et al., 2004*). Interestingly, deleting or substituting the tuning receptor for an ORN does not alter its characteristic spike amplitude (Dobritsa, *van der Goes van Naters and Carlson, 2007*; *Hallem et al., 2004*), suggesting that spike amplitude is a feature independent of receptor identity. The genetically predetermined pairing of neurons implies functional significance for this arrangement.

Indeed, ORNs housed in the same sensillum can inhibit each other by means of direct electrical interaction, termed ephaptic coupling, which can also modulate fruitfly behavior in response to odor mixtures (*Su et al., 2012*; *Zhang et al., 2019*). Strikingly, in most sensillum types, lateral inhibition is asymmetric between compartmentalized ORNs: the large-spike neuron is not only capable of exerting greater ephaptic influence but is also less susceptible to ephaptic inhibition by its small-spike neighbors. Mechanistically, this functional disparity arises from the size difference between grouped neurons. The large-spike ORN has a larger soma than its small-spike neighbor(s); this feature is translated into a smaller input resistance for the 'A' neuron, thus accounting for its dominance in ephaptic interaction (*Zhang et al., 2019*). These findings indicate a causal relationship between an ORN's morphometric features and its circuit function. Taken together, these observations highlight the necessity and value of a systematic morphometric survey of genetically identified ORNs.

Pioneering transmission electron microscopy (TEM) studies of high-pressure frozen and freeze-substituted (i.e., cryofixed) antennal tissues have provided a high-quality morphological atlas of *Drosophila* ORNs (*Shanbhag et al., 1999*; *Shanbhag et al., 2000*). However, for morphometric analysis, the utility of this atlas is limited because the 3D structure and molecular identity of ORNs are undefined. Taking advantage of the CryoChem method, which we have previously developed to permit high-quality ultrastructural preservation of cryofixed and genetically labeled samples for volume EM (*Tsang et al., 2018*), we have acquired serial block-face scanning electron microscopy (SBEM) images of antennal tissues in which select ORNs expressed a membrane-tethered EM marker (APEX2-mCD8GFP or APEX2-ORCO) (*Tsang et al., 2018*; *Zhang et al., 2019*). In a previous study, we focused our analysis on the soma of labeled ORNs and their co-compartmentalized neuronal partners in five sensillum types (*Zhang et al., 2019*). Yet, in addition to the APEX2-labeled hairs, each SBEM data set contains many other sensilla, whose identity can be determined by cross-referencing the sensillum morphology and the number of its compartmentalized ORNs (*Benton et al., 2009*; *Couto et al., 2005*; *Fishilevich and Vosshall, 2005*; *Grabe et al., 2016*; *Lin and Potter, 2015*; *Shanbhag et al., 1999*, *Shanbhag et al., 2000*). For example, among the three large basiconic sensillum types, ab1 can be identified without genetic labeling because the sensillum houses four neurons, whereas ab2 and ab3 both house two ORNs. Moreover, within a known sensillum, the 'A' or 'B' ORN can be distinguished according to the soma size. That is, the spike amplitude ratio between grouped neurons reflects their soma size differential (*Zhang et al., 2019*). Thus, the cellular identities of ORNs can be determined for many unlabeled neurons in our SBEM data sets (see *Table 1—source data 1* for details).

Here, we conduct a systematic morphological and morphometric analysis of identified *Drosophila* ORNs. By segmenting SBEM images, we reconstruct 3D structures for neurons housed in 13 of the 19 genetically defined antennal sensillum types (*Benton et al., 2009*; *Lin and Potter, 2015*), covering all four morphological classes: the basiconic (ab1–5), coeloconic (ac1–4), intermediate (ai2 and ai3), and trichoid sensilla (at1 and at4) from a total of 158 reconstructed 3D sensillum models. Furthermore, based on 122 reconstructed 3D ORN models, we perform morphometric analysis for 33 of the 40 identified antennal ORN types (*Task and Lin, 2021*), as well as for 4 novel neuronal types. Our high-resolution survey uncovers multiple novel features, including extracellular vacuoles

**Table 1.** Morphometric information on identified olfactory sensilla of *Drosophila melanogaster*.

Values are means ± s.e.m. Empty sensilla which do not house any neurons are denoted by (0). abx: novel basiconic sensillum, and the numbers of compartmentalized neurons are indicated in parentheses. Detailed information regarding how individual sensilla were identified and their respective source data sets can be found in the *Table 1—source data 1* and *Figure 1—source data 1* files.

| Sensillum type | Length (μm) | Surface area (μm$^2$) | Cuticle volume (μm$^3$) | Lumen volume (μm$^3$) |
|---|---|---|---|---|
| *Trichoid* | | | | |
| T1 | 21.29±0.35 (n=19) | 103.33±1.63 | 44.70±0.75 | 22.74±0.41 |
| T2 | 21.94±0.41 (n=10) | 116.75±1.75 | 55.73±0.96 | 28.19±1.03 |
| T3 | 20.64±0.27 (n=21) | 112.94±1.39 | 55.58±0.85 | 28.45±0.51 |
| T(0) | 22.13 (n=1) | 103.60 | 47.98 | 24.38 |
| *Intermediate* | | | | |
| ai2 | 12.70±0.19 (n=20) | 55.26±1.02 | 21.92±0.50 | 11.08±0.26 |
| ai3 | 10.97±0.12 (n=20) | 52.49±1.12 | 21.38±0.66 | 11.83±0.25 |
| *Basiconic* | | | | |
| ab1 | 10.80±0.16 (n=5) | 70.70±0.55 | 39.49±2.12 | 28.18±2.38 |
| ab2 | 10.33±0.30 (n=4) | 62.86±3.63 | 32.28±2.82 | 23.18±1.87 |
| ab3 | 11.92±0.27 (n=7) | 77.06±3.66 | 38.72±1.47 | 30.03±1.07 |
| abx(3) | 10.37±0.11 (n=2) | 71.48±2.38 | 36.49±0.06 | 23.79±0.34 |
| ab4 | 11.31±0.15 (n=7) | 63.28±2.98 | 27.84±1.87 | 17.41±1.84 |
| ab5 | 8.84±0.19 (n=8) | 36.85±0.68 | 13.49±0.30 | 6.85±0.22 |
| abx(1) | 8.84±0.16 (n=3) | 42.25±1.48 | 16.18±1.77 | 8.68±1.44 |
| abx(0) | 9.00 (n=1) | 39.00 | 15.09 | 7.30 |
| *Coeloconic* | | | | |
| ac1 | 6.19±0.17 (n=18) | 32.40±1.36 | 13.51±0.73 | 0.44±0.04 |
| ac2 | 5.68±0.06 (n=3) | 21.59±1.33 | 8.98±0.33 | 0.40±0.03 |
| ac3II | 5.39±0.13 (n=8) | 25.16±0.59 | 9.86±0.42 | 0.82±0.04 |
| ac4 | 4.89 (n=1) | 26.52 | 9.14 | 0.32 |

The online version of this article includes the following source data for Table 1:

**Source data 1.** Source data for *Table 1*.

in the sensillum lumen, mitochondria-enriched inner dendritic enlargement, empty sensilla which house no ORNs, and two hitherto uncharacterized basiconic sensillum types. We also show that the outer dendritic branching pattern is much more intricate than previously appreciated. The information provided here defines new questions about how the size and shape of sensory neurons influence their responses, sensitivity, and circuit function, and sets the stage for future investigations into the molecular and cellular mechanisms underlying diverse ORN morphologies.

## Results and discussion

We began with a detailed survey of the SBEM images we had previously generated using female antennal tissues (*Zhang et al., 2019*). Individual data sets, acquired from different flies and named on the basis of the APEX2-expressing ORNs, included numerous sensory hairs in addition to the sensilla that housed labeled neurons (*Figure 1A*). In agreement with the characteristic topographical distribution of sensilla on the antenna (*de Bruyne et al., 2001*; *Grabe et al., 2016*; *Shanbhag et al., 1999*), the four morphological sensillum classes were unevenly represented in our eight SBEM data sets (*Figure 1B,C*). A total of 541 sensilla were sampled from these data sets, comparable to the total number of sensilla on an antenna (*Shanbhag et al., 1999*). Among the whole sensilla which were sampled in their entirety, we identified 37 basiconic, 30 coeloconic, 40 intermediate, and 51 trichoid sensilla based on their distinctive morphological features (*Shanbhag et al., 1999*). Briefly, the cuticle of basiconic, intermediate, and trichoid sensilla is decorated with numerous pores which serve

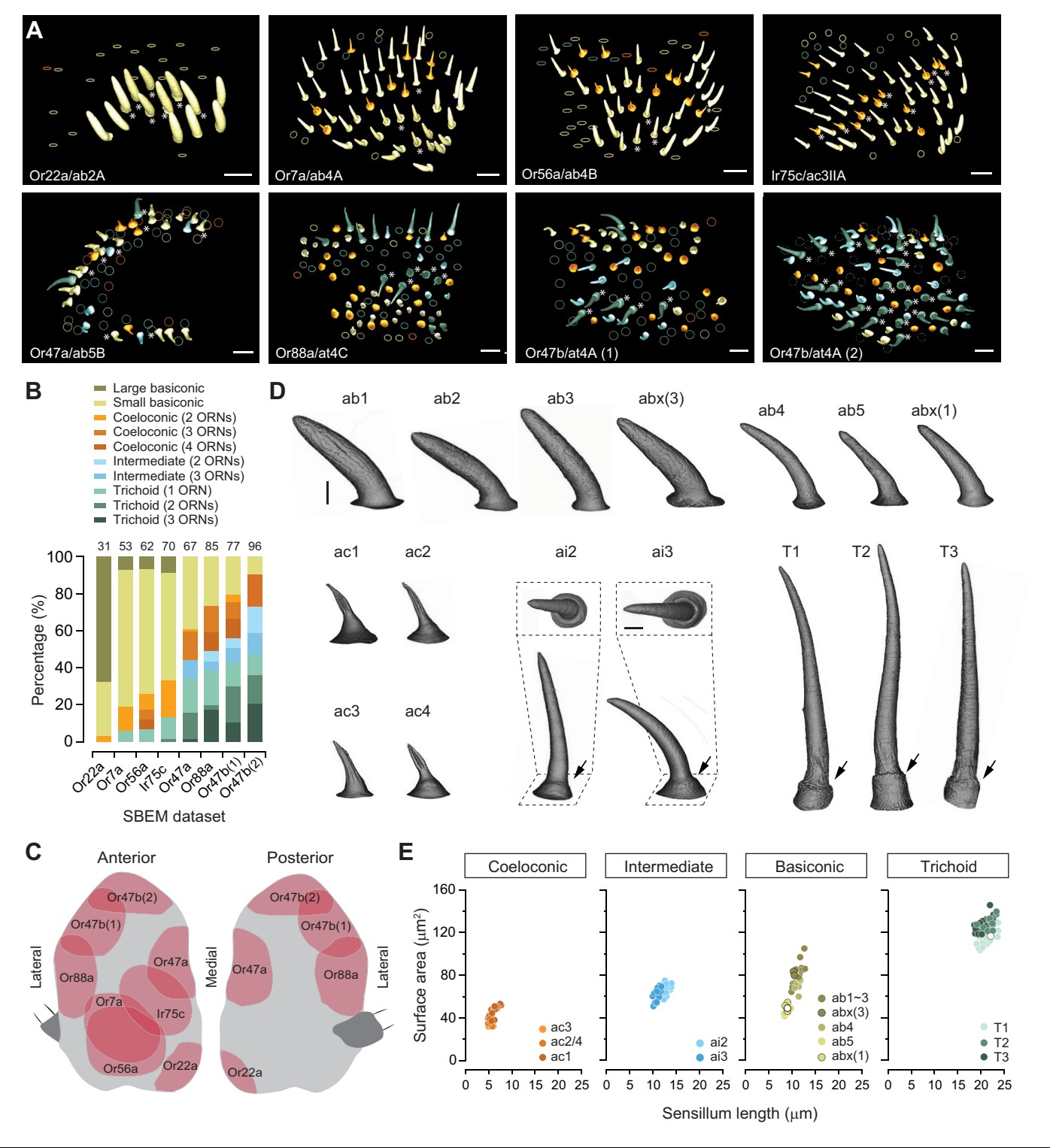

**Figure 1.** Different sensillum classes and their distribution across SBEM data sets. (**A**) SBEM data sets analyzed in this study, named according to the genetically labeled ORNs (lower left corner, the receptor, and ORN identity are indicated). The 3D models of the cuticles are shown for basiconic (yellow), coeloconic (orange), intermediate (light blue), and trichoid sensilla (teal). Asterisks indicate the sensilla containing genetically labeled ORNs. Circles indicate partially sampled sensilla. The data sets are arranged approximately along the proximal-medial to distal-lateral axis on the anterior face of an antenna. Scale bars: 10 μm. (**B**) Percentage of different sensillum classes in individual data sets. The total numbers of sampled sensilla are indicated above the graph. (**C**) Illustration of antennal regions covered by individual SBEM data sets. (**D**) Isosurface images of identified sensillum types.

*Figure 1 continued on next page*

*Figure 1 continued*

Large basiconic: ab1, ab2, ab3, and abx(3); thin basiconic: ab4; and small basiconic: ab5 and abx(1). Coeloconic: ac1, ac2, ac3, and ac4. Intermediate: ai2 and ai3; insets: top-down perspective. Trichoid: T1, T2, and T3. Arrows indicate the basal drum. Scale bars: 2 μm. (**E**) Different sensillum classes exhibit distinctive morphometric features. The surface area for completely segmented sensillum cuticle is plotted as a function of sensillum length. Empty circles indicate the sensilla which do not house any ORNs. ORN, olfactory receptor neuron; SBEM, serial block-face scanning electron microscopy.

The online version of this article includes the following source data for figure 1:

**Source data 1.** Source data for *Figure 1*.

as passages for odorants. The pores of basiconic sensilla are arranged in characteristic arrays, whereas those of trichoid and intermediate sensilla appear to scatter irregularly on the cuticular surface. The trichoid and intermediate sensilla are further recognized by their cylindrical, thickened bases, termed the 'basal drum' (*Figure 1D*). In contrast, the cuticle of the coeloconic sensilla does not contain any pores but instead consists of multiple cuticular fingers (*Figure 1D*), between which numerous spoke channels allow odorants access to ORNs (*Shanbhag et al., 1999*).

For the completely sampled sensilla, we reconstructed their 3D models and performed morphometric analysis. Within a morphological class, sensillum identity was determined by the number of enclosed neurons and by genetic labeling when this information was available. Of note, the distribution of individual sensillum types on the antenna follows a characteristic topographical pattern (*de Bruyne et al., 2001*; *Grabe et al., 2016*; *Shanbhag et al., 1999*). Therefore, for an unlabeled sensillum, its proximity to the labeled sensillum type may further aid its identification (see Large basiconic sensillum section below). For basiconic sensilla, we identified all three large basiconic sensilla (ab1, ab2, and ab3), one thin basiconic (ab4), and one small basiconic sensillum (ab5). Unexpectedly, we also found two novel types of basiconic sensilla: one large basiconic housing three neurons, which we termed abx(3), and the other, a small basiconic housing only one neuron, which we termed abx(1). For the other three morphological sensillum classes, all of the reported subtypes were identified, including four coeloconic, two intermediate, and three trichoid sensilla (*Figure 1D*).

We further measured the length, surface area, cuticular, and lumen volumes of these identified sensilla (*Table 1*). The coeloconic sensilla are the shortest (5–8 μm), followed by the ab5 small basiconic (8–10 μm). The intermediate, thin, and large basiconic sensilla are of similar length (10–14 μm), while the trichoid sensilla are the longest (19–24 μm) (*Figure 1E*). In the remainder of this article, we focus on individual sensillum types, describe their identification, and provide morphological and morphometric information for their neurons (see Source data files for detailed information).

## Trichoid sensilla

### T1 sensillum

Distinctly among trichoids, the T1 sensillum houses only a single neuron expressing the Or67d receptor that responds to the male fly pheromone, cVA (*Couto et al., 2005*; *Kurtovic et al., 2007*; *Shanbhag et al., 1999*; *Ha and Smith, 2006*; *van der Goes van Naters and Carlson, 2007*). The ORN soma is located approximately 8–16 μm beneath the antennal surface (*Figure 2A* and *Figure 2—figure supplement 1*), with an average size of 47 μm$^3$ (*Table 2*). The dendrite is divided into the inner and outer segments by the ciliary constriction (*Figure 2A*; *Shanbhag et al., 1999*). Situated entirely beneath the antennal surface, the inner dendritic segment is ensheathed by the thecogen cell, which is itself enveloped by the trichogen processes and then further surrounded by the tormogen cell (*Figures 2A–7* and *8*). The inner dendrite exhibits a wider diameter (0.75–1.5 μm) than the outer dendrite (~0.3 μm), which extends into the sensillum lumen with a slight taper (*Figure 2A*) or with a markedly flattened distal segment (*Figure 2—figure supplement 1A* and *Video 1*). Although the T1 outer dendrite is believed to be typically unbranched (*Shanbhag et al., 1999*), we observed that among the four segmented T1 ORNs, one exhibits three branches and another six (*Figure 2—figure supplement 1B,C* and *Video 1*). In both cases, the dendrites began branching approximately halfway through the dendritic length inside the lumen, with secondary branching occurring further along distally (*Figure 2—figure supplement 1B,C*). The outer dendrite typically extends to the tip of the sensillum (*Figure 2—figure supplement 1*), but some terminate before reaching the endpoint (*Figure 2A*).

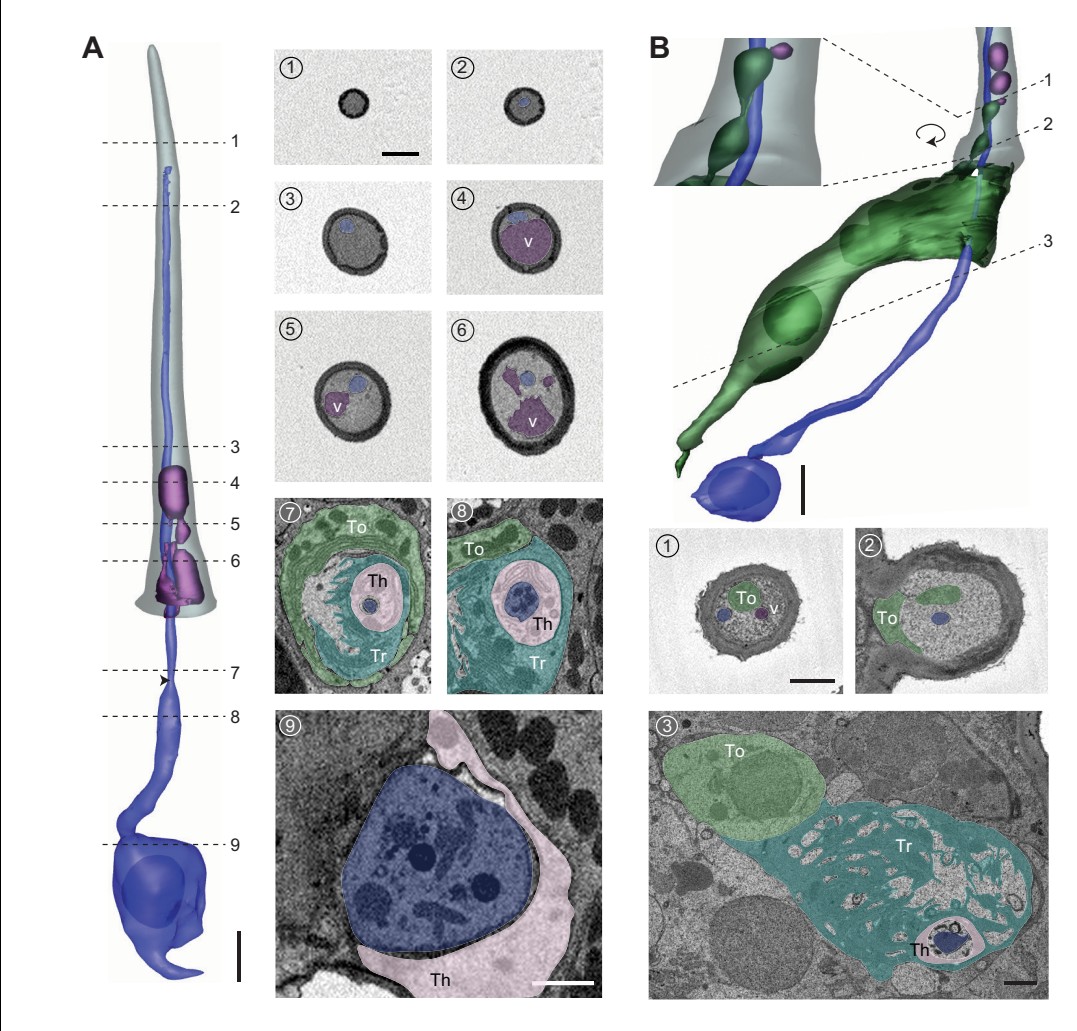

**Figure 2.** T1 trichoid sensillum. (**A**) 3D model and SBEM images of a representative T1 sensillum. ORN (pseudocolored in blue), vacuoles (aubergine), and sensillum cuticle (gray) are shown. Dashed lines indicate positions of the corresponding SBEM images on the right: (1) luminal space devoid of ORN dendrite; (2–3) outer dendrite; (4–6) outer dendrite and vacuole structures; (7) ciliary constriction (arrow head), which demarcates the inner and outer dendritic segments; (8) inner dendrite; and (9) ORN soma. Three types of auxiliary cells were observed beneath the antennal surface: thecogen (Th), trichogen (Tr), and tormogen (To). (**B**) A partial 3D model and SBEM images of another T1 sensillum. ORN (blue), vacuoles (aubergine), and tormogen cell (green) are shown. Inset: magnified and rotated view, highlighting the bulbous protrusion from the tormogen cell. Dashed lines indicate positions of the corresponding SBEM images below: (1) vacuole in close proximity to the tormogen protrusion; (2) tormogen cell and its protrusion; and (3) tormogen soma, trichogen, and thecogen processes, and ORN dendrite. Scale bars: 2 μm for 3D models and 1 μm for SBEM images. The scale bar in the first image panel also pertains to other images unless indicated otherwise. ORN, olfactory receptor neuron; SBEM, serial block-face scanning electron microscopy.

The online version of this article includes the following figure supplement(s) for figure 2:

**Figure supplement 1.** Diverse T1 dendritic morphologies.

Besides the dendritic branches, we observed free, extracellular membrane-bound structures within the lumen in 4 of the 19 analyzed T1 sensilla (21%) (see *Table 1—source data 1* for details). These membranous structures, which we referred to as vacuoles, were predominantly found in the proximal region of the sensillum (*Figure 2A-4-6*). They vary considerably in size (0.2–2.0 μm in diameter) and do not appear to have a regular shape (*Figure 2A*). In some cases, the vacuole occupied nearly an entire segment of the lumen, reducing the ORN dendrite to the margin (*Figures 2A–*

**Table 2.** Morphometric information on identified ORNs of *Drosophila melanogaster*.

ORN identity is indicated by the sensillum type, relative spike amplitude (A, B, C, or D), odor-tuning receptor, and glomerular projection. Values are means ± s.e.m. Mitochondria analysis was performed for select ORN types. abx: novel basiconic sensillum, and the numbers of compartmentalized neurons are indicated in parentheses. * Information is available for one model. # Typical ac1 neurons. NA: information is not available because of incomplete sampling or segmentation. The odor response profiles for many of the characterized ORNs can be found in the DoOR database (http://neuro.uni-konstanz.de/DoOR/default.html).

| ORN identity | | Soma | | Inner dendrite | | | Outer dendrite | | |
|---|---|---|---|---|---|---|---|---|---|
| | | Volume (µm³) | Surface area (µm²) | Volume (µm³) | Surface area (µm²) | Mitochondria number | Volume (µm³) | Surface area (µm²) | Branch number |
| T1 (n=4) | A (Or67d) (DA1) | 47.44±0.95 | 79.21±2.25 | 5.08±0.20 | 31.80±2.14 | NA | 2.58±0.21 | 36.70±3.93 | 2.75±1.18 |
| T2 (at4) (n=4) | A (Or47b) (VA1v) | 63.42±5.41 | 92.70±4.54 | 6.50±0.20 | 37.62±1.42 | NA | 2.40±0.21 | 26.98±1.57 | 1.00±0.00 |
| | B (Or88a) (VA1d) | 34.90±1.04 | 65.07±1.44 | 4.50±0.61 | 29.70±2.44 | NA | 1.00±0.05 | 19.99±1.27 | 2.00±0.41 |
| T3 (at4) (n=8) | A (Or47b) (VA1v) | 67.10±3.75 | 100.19±4.09 | 7.28±0.46 | 39.04±2.84 | NA | 2.75±0.38 | 30.55±2.85 | 1.57±0.27 |
| | B (Or65a) (DL3) | 26.15±1.18 | 54.01±1.65 | 3.57±0.60 | 26.53±2.72 | NA | 1.02±0.33 | 13.46±1.64 | 1.25±0.25 |
| | C (Or88a) (VA1d) | 44.35±3.63 | 83.73±4.77 | 6.10±0.57 | 37.23±2.92 | NA | 1.59±0.44 | 23.47±2.85 | 1.50±0.38 |
| ai2 (n=3) | A (Or83c) (DC3) | 47.96±2.95 | 78.43±2.85 | 7.23±0.30 | 29.28±1.77 | NA | 2.92±0.17 | 41.85±2.44 | 7.67±1.76 |
| | B (Or23a) (DA3) | 30.22±2.06 | 62.92±0.68 | 3.07±0.08 | 19.69±1.85 | NA | 0.64±0.07 | 13.18±1.63 | 3.33±0.33 |
| ai3 (n=2) | A (Or19a) (DC1) | 64.05±2.45 | 108.63±6.88 | 8.78±0.84 | 31.71±3.96 | NA | 2.51±0.12 | 35.85±2.15 | 16.00±2.00 |
| | B (Or2a) (DA4m) | 43.65±1.30 | 78.26±3.49 | 6.26±1.00 | 31.16±5.55 | NA | 1.46±0.09 | 29.08±2.29 | 15.5±0.50 |
| | C (Or43a) (DA4l) | 36.72±5.21 | 71.48±5.45 | 4.44±1.16 | 22.66±5.21 | NA | 1.03±0.14 | 19.08±2.29 | 9.00±0.00 |
| ab1 (n=2) | A(Or42b) (DM1) | 69.96±4.14 | 117.76±1.55 | 19.39±4.63 | 63.22±7.02 | 123.50±14.50 | NA | NA | NA |
| | B (Or92a) (VA2) | 60.96±4.11 | 98.22±3.89 | 21.15±0.83 | 70.57±11.40 | 81.50±21.50 | NA | NA | NA |
| | C (Gr21a/63a) (V) | 52.10±1.00 | 105.23±0.48 | 6.73±0.63 | 29.84±1.20 | 26.00±2.00 | NA | NA | NA |
| | D (Or10a) (DL1) | 28.03±1.69 | 61.00±1.17 | 2.83±0.01 | 19.92±1.53 | 9.00±3.00 | NA | NA | NA |
| abx(3) (n=3) | A | 59.20±5.98 | 125.35±5.38 | 17.49±6.17 | 74.81±9.04 | NA | NA | NA | NA |
| | B | 53.79±2.83 | 126.34±8.41 | 18.41±2.68 | 89.24±23.85 | NA | NA | NA | NA |
| | C | 43.71±0.88 | 109.37±13.37 | 5.90±0.02 | 43.71±3.72 | NA | NA | NA | NA |
| ab2 (n=1) | A (Or59b) (DM4) | 93.69 | 131.28 | 20.01 | 63.01 | 125 | 9.8 | 275.42 | 312 |
| | B (Or85a) (DM5) | 34.40 | 68.89 | 3.32 | 26.73 | 6 | 0.49 | 13.78 | 1 |
| ab3 (n=5) | A (Or22a) (DM2) | 50.35±2.42 | 98.27±3.74 | 13.42±1.17 | 65.64±5.33 | 75* | 6.15* | 144.73* | 83* |
| | B (Or85b) (VM5d) | 32.97±1.49 | 87.30±6.68 | 3.69±0.42 | 26.92±3.46 | 14* | 1.95* | 39.55* | 19* |
| ab4 (n=3) | A (Or7a) (VM5v) | 73.44±3.38 | 101.94±3.10 | 9.44±0.63 | 32.60±2.91 | 62.33±6.01 | 3.90±0.42 | 79.89±18.34 | 46.67±20.79 |
| | B (Or56a) (DA2) | 35.22±1.12 | 67.72±4.01 | 3.63±0.45 | 18.89±2.49 | 17.67±4.48 | 1.65±0.40 | 30.79±3.74 | 12.67±4.10 |
| ab5 (n=4) | A (Or82a) (VA6) | 31.21±1.16 | 61.53±3.02 | 3.77±0.34 | 19.79±2.05 | NA | 0.98±0.05 | 27.25±2.18 | 39.25±8.78 |
| | B (Or47a) (DM3) | 41.11±3.70 | 72.45±4.73 | 1.77±0.45 | 14.70±2.17 | NA | 0.82±0.16 | 23.77±4.77 | 44.75±12.38 |
| abx(1) (n=3) | A | 38.42* | 89.88* | 7.00±0.74 | 36.09±3.85 | 38.50±1.50 | 1.28±0.02 | 20.94±2.75 | 6.00±3.00 |
| ac1# (n=4) | A (Rh50/Amt) (VM6) | 23.20±0.55 | 46.98±0.92 | 1.38±0.08 | 9.96±0.87 | 4.75±0.48 | 0.62±0.04 | 7.95±0.43 | 1.00±0.00 |
| | B (Ir92a) (VM1) | 11.56±0.40 | 32.02±1.29 | 0.84±0.01 | 8.47±0.27 | 3.50±0.96 | 0.13±0.01 | 3.72±0.15 | 1.00±0.00 |
| | C (Ir31a or 75d) (VL2p or VL1) | 9.62±0.33 | 26.03±0.62 | 0.54±0.03 | 5.29±0.20 | 1.50±0.50 | 0.13±0.01 | 3.83±0.11 | 1.00±0.00 |
| | D (Ir31a or 75d) (VL2p or VL1) | 7.92±0.35 | 25.08±1.50 | 0.67±0.04 | 8.54±0.92 | 2.25±0.63 | 0.20±0.01 | 4.65±0.13 | 1.00±0.00 |
| ac2 (n=3) | A (Ir75a) (DP1l) | 26.69±1.48 | 64.46±4.81 | 3.08±1.00 | 20.41±5.89 | NA | 0.36±0.02 | 7.36±0.32 | 1.00±0.00 |
| | B (Ir75d or Ir41a) (VL1 or VC5) | 24.05±0.81 | 54.33±1.48 | 2.07± 0.06 | 14.87±0.85 | NA | 0.48±0.05 | 8.59±0.34 | 1.00±0.00 |
| | C (Ir75d or Ir41a) (VL1 or VC5) | 15.42±0.58 | 40.32±4.12 | 1.55±0.22 | 11.54±1.36 | NA | 0.26±0.05 | 6.41±0.73 | 1.00±0.00 |

*Table 2 continued on next page*

*Table 2 continued*

| ORN identity | | Soma | | Inner dendrite | | | Outer dendrite | | |
|---|---|---|---|---|---|---|---|---|---|
| | | Volume (μm³) | Surface area (μm²) | Volume (μm³) | Surface area (μm²) | Mitochondria number | Volume (μm³) | Surface area (μm²) | Branch number |
| ac3II (n=4) | A (Ir75c) (DL2) | 41.27±5.65 | 99.52±6.22 | 3.75±0.36 | 26.53±1.84 | NA | 0.91±0.07 | 12.18±0.49 | 1.25±0.25 |
| | B (Or35a) (VC3) | 23.49±1.30 | 60.44±3.70 | 3.02±0.43 | 24.96±2.47 | NA | 0.59±0.11 | 11.01±1.15 | 3.00±0.71 |
| ac4 (n=1) | A (Ir84a) (VL2a) | 38.30 | 64.62 | 1.99 | 11.76 | NA | 0.24 | 5.17 | 1.00 |
| | B (Ir75d) (VL1) | 31.99 | 59.07 | 2.72 | 15.89 | NA | 0.47 | 7.44 | 1.00 |
| | C (76a) (VM4) | 17.92 | 39.60 | 1.62 | 10.65 | NA | 0.25 | 5.01 | 1.00 |

The online version of this article includes the following source data for Table 2:

**Source data 1.** Source data for *Table 2*.

*4*). Although their functional significance remains to be determined, the extracellular vacuoles are likely derived from the tormogen auxiliary cell. In one T1 sensillum, we found that its tormogen cell extended a bulbous protrusion into the lumen, resembling the process of membrane budding (*Figure 2B*). Immediately adjacent to the bulbous protrusion was a small vacuole (*Figure 2B*). The close proximity between these two structures suggests that the free vacuoles within the lumen may have originated from the tormogen cell.

## T2 and T3 sensilla

In addition to T1, we identified the other two trichoid sensillum types—T2 and T3—containing two and three neurons, respectively (*Shanbhag et al., 1999*; *Figure 1*). All trichoid sensilla are similar in length (19–24 μm), but T1 is distinctively narrower than T2 or T3 with smaller sensillum surface area and volume (*Figure 1C,D* and *Table 1*). In contrast, T2 and T3 cannot be distinguished externally from their morphological or morphometric features (*Figure 1C,D* and *Table 1*). On the basis of recent molecular and functional studies, T3 likely corresponds to the at4 sensillum, which houses three pheromone-sensing neurons: the Or47b (at4A), Or65a (at4B), and Or88a ORNs (at4C) (*Couto et al., 2005*; *Dweck et al., 2015*; *Lin et al., 2016*; *van der Goes van Naters and Carlson, 2007*). However, the molecular identity for T2 neurons remains hitherto unknown. Unexpectedly, in the data sets in which Or47b ORNs were genetically marked (*Or47b-GAL4>10XUAS-myc-APEX2-Orco*), we identified labeled neurons in both T2 and T3 sensilla (*Figure 3A,B*). Similarly, in the Or88a data set (*Or88a-GAL4>10XUAS-myc-APEX2-Orco*), labeled neurons were found in both T2 and T3 sensilla (*Figure 3—figure supplement 1*). Taken together, these results indicate that Or47b and Or88a ORNs are the shared constituents for T2 and T3.

In both sensillum types, the at4A/Or47b ORN is the largest cell among compartmentalized neurons, measuring approximately 65 μm³ in soma volume. In comparison, the at4C/Or88a soma is smaller, averaging 35 μm³ in T2 and 44 μm³ in T3. Although the Or65a ORN was previously designated as at4B without clear evidence for its intermediate spike amplitude, its soma is in fact the smallest among the three at4 ORNs, averaging only 26 μm³ (*Figure 3C* and *Table 2*). As with soma size, the average inner and outer dendritic volumes are both greatest for at4A/Or47b neurons, followed by at4C/Or88a, and then at4B/Or65a ORNs (*Figure 3D,E*). However, compared to soma size, a greater variability was observed for dendritic volumes. For example, in one T3 sensillum, the at4B/Or65a outer dendrite was larger than that of the neighboring at4C/Or88a neuron; in other instances, the at4C/Or88a dendritic volume was greater than that of at4A/Or47b (*Figure 3E*).

Similar to the T1 neurons, the inner dendritic segments of T2 or T3 ORNs are insulated from one another by thecogen processes (*Figures 3A-3 and 4* and *Figures 3B-3 and 4*). The outer dendrites of some T2 and T3 ORNs also exhibit a small number of branches: among 12 pairs of segmented neurons, three Or47b ORNs display two or three dendritic branches, and five Or88a neurons have two to four branches. In comparison, only one of the eight segmented Or65a ORNs exhibit branched outer dendrites (*Table 2* and *Table 2—source data 1* for details). Also similar to T1 neurons, the outer dendritic morphology of T2 and T3 neurons are typically cylindrical with a gradual taper toward the tip, while some neurons have portions of their outer dendritic segments flattened like a rippled ribbon (*Figure 3—figure supplement 2A,B* and *Video 2*). Notably, we observed ring-

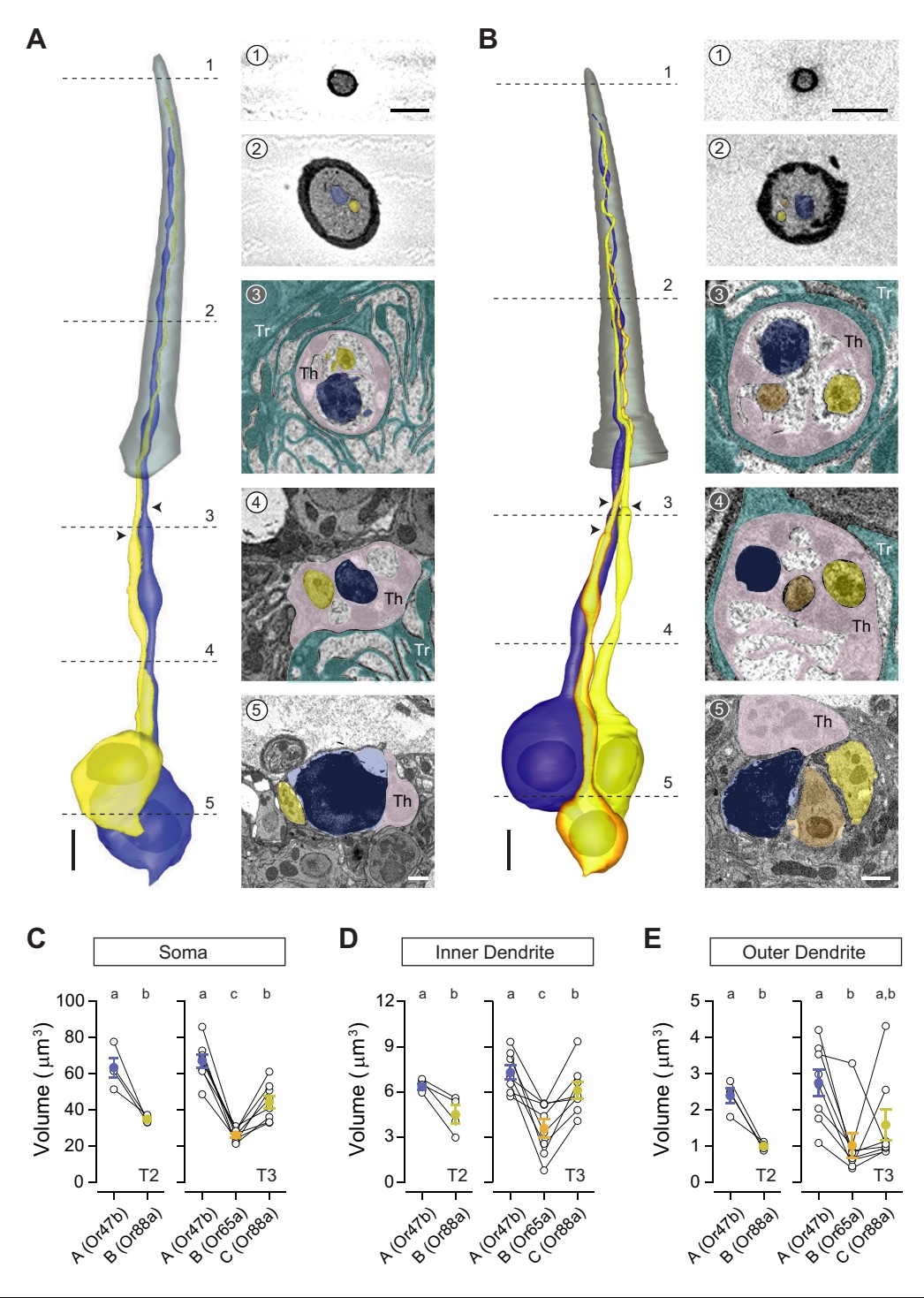

**Figure 3.** T2 and T3 trichoid sensilla. 3D models and SBEM images of a representative T2 (**A**) or T3 sensillum (**B**). Images were acquired from transgenic flies in which Or47b ORNs were engineered to express APEX2 (*Or47b-GAL4>10XUAS-myc-APEX2-Orco*) and labeled with diaminobenzidine (DAB, darker staining). (**A**) T2 houses two ORNs: the larger neuron is DAB-labeled and pseudocolored in blue, whereas the unlabeled, smaller neighbor is pseudocolored in yellow. (**B**) T3 houses three ORNs of different sizes: large (DAB-labeled and pseudocolored in blue), medium (yellow), and small (orange). Dashed lines indicate positions of the corresponding SBEM images on the right. Arrow heads indicate ciliary constrictions. Th: thecogen; Tr: trichogen; and To: tormogen. Scale bars: 2 μm for 3D models and 1 μm for SBEM images. The scale bar in the first image panel also pertains to other images unless indicated otherwise. (**C–E**) Volumes of the soma (**C**), inner (**D**), and outer dendrites (**E**) of T2 or T3 ORNs.
*Figure 3 continued on next page*

*Figure 3 continued*

Lines connect data points from neurons housed in the same sensillum, mean± s.e.m. n=4 for T2, and n=8 for T3. Significant differences (p<0.05) are indicated by different letters; paired t-test for comparison between neighboring neurons, and unpaired t-test for comparison between non-grouped neurons. ORN, olfactory receptor neuron; SBEM, serial block-face scanning electron microscopy.

The online version of this article includes the following figure supplement(s) for figure 3:

**Figure supplement 1.** Or88a ORN is identified in both T2 and T3 sensilla.

**Figure supplement 2.** Diverse dendritic morphologies of Or47b and Or88a ORNs.

shaped outer dendritic profiles in the cross-section images of some unbranched Or47b ORNs; 3D reconstruction reveals that these profiles correspond to dendritic regions with a hollow center (*Figure 3—figure supplement 2C,D*). In one neuron, the hollowed region was short, observed in the mid-section of the outer dendrite (*Figure 3—figure supplement 2C*). In another neuron, the hollowed section was considerably longer, exceeding 1/3 of the outer dendritic length, and was found in the distal segment of the dendrite (*Figure 3—figure supplement 2D*). Their variable position and length suggest that this dendritic structure is dynamic, possibly reflecting the process of dendritic branching. Finally, as with T1 sensilla, we observed extracellular vacuoles in the proximal regions of the sensillum lumen in 27% of the analyzed T2 sensilla and in 29% of T3 sensilla (see *Table 1—source data 1* for details).

## Coeloconic sensilla

Conical in shape, the coeloconic sensilla are the shortest antennal sensilla, ranging from 4.8 to 7.6 µm long (*Figure 1C,D*). Unlike other olfactory sensilla, the cuticle of the coeloconic sensilla contains no pores and instead consists of closely apposed cuticular fingers, between which longitudinal grooves permit odorants to enter the inner sensillum lumen through interspersed spoke channels (*Shanbhag et al., 1999*). The number of cuticular fingers ranges from 5 to 9, and does not correlate with sensillum types (see *Table 1—source data 1* for details). The cuticle is double-walled, wherein the inner cuticular cylinder lines the inner lumen, the volume of which is the smallest among antennal sensilla (0.3–1.0 µm$^3$) (*Table 1*). Extracellular vacuoles were not typically observed within the small luminal volumes, as vacuoles were found only in 6 of the 90 analyzed coeloconic sensilla (7%) (see *Table 1—source data 1* for details). The inner sensillum lumen encapsulates only a short distal segment of the outer dendrite (2–3 µm), while approximately 75% of the outer dendritic length is beneath the antennal surface. In addition, the coeloconic outer dendrites typically terminate before reaching the sensillum tip (*Figure 4* and *Video 3*).

## ac3 sensillum

Based on the number of compartmentalized neurons, the coeloconic sensilla are classified into three types that house two, three, or four ORNs. Among the two-neuron coeloconic sensilla, ac3I and ac3II are distinguished by genetic labeling. While the Or35a receptor is common to the B neurons in both types (*Yao et al., 2005*), the ac3IA neuron expresses the Ir75b receptor, whereas ac3IIA expresses Ir75c (*Prieto-Godino et al., 2017*). Here, we focused our analysis on the ac3II neurons identified from the Ir75c-labeled data set. The ac3II sensilla share similar external morphometric features as other coeloconic sensilla. However, they have the largest inner sensillum lumen, averaging 0.82 µm$^3$, as opposed to 0.32–0.44 µm$^3$ for other coeloconics (*Table 1*). Toward the distal end of the sensillum, ac3II has a flower-shaped cross-section profile due to its round cuticular fingers (*Figure 4A*). Inside the sensillum, the soma, inner, and outer dendrites of ac3IIA are consistently larger than those of ac3IIB (*Figure 4A* and *Table 2*). Among four pairs of segmented ac3II neurons, one ac3A has two dendritic branches, while all ac3B neurons have branched outer dendrites (2–5 branches, *Table 2* and *Video 3*).

## ac1 sensillum

A recent study shows that the ac1 sensillum compartmentalizes four neurons, including an ORN that expresses the ammonium transporter Amt as an ammonia receptor (*Vulpe et al., 2021*). The other three ac1 neurons express the Ir92, Ir31a, and Ir75d receptors, respectively (*Benton et al., 2009*).

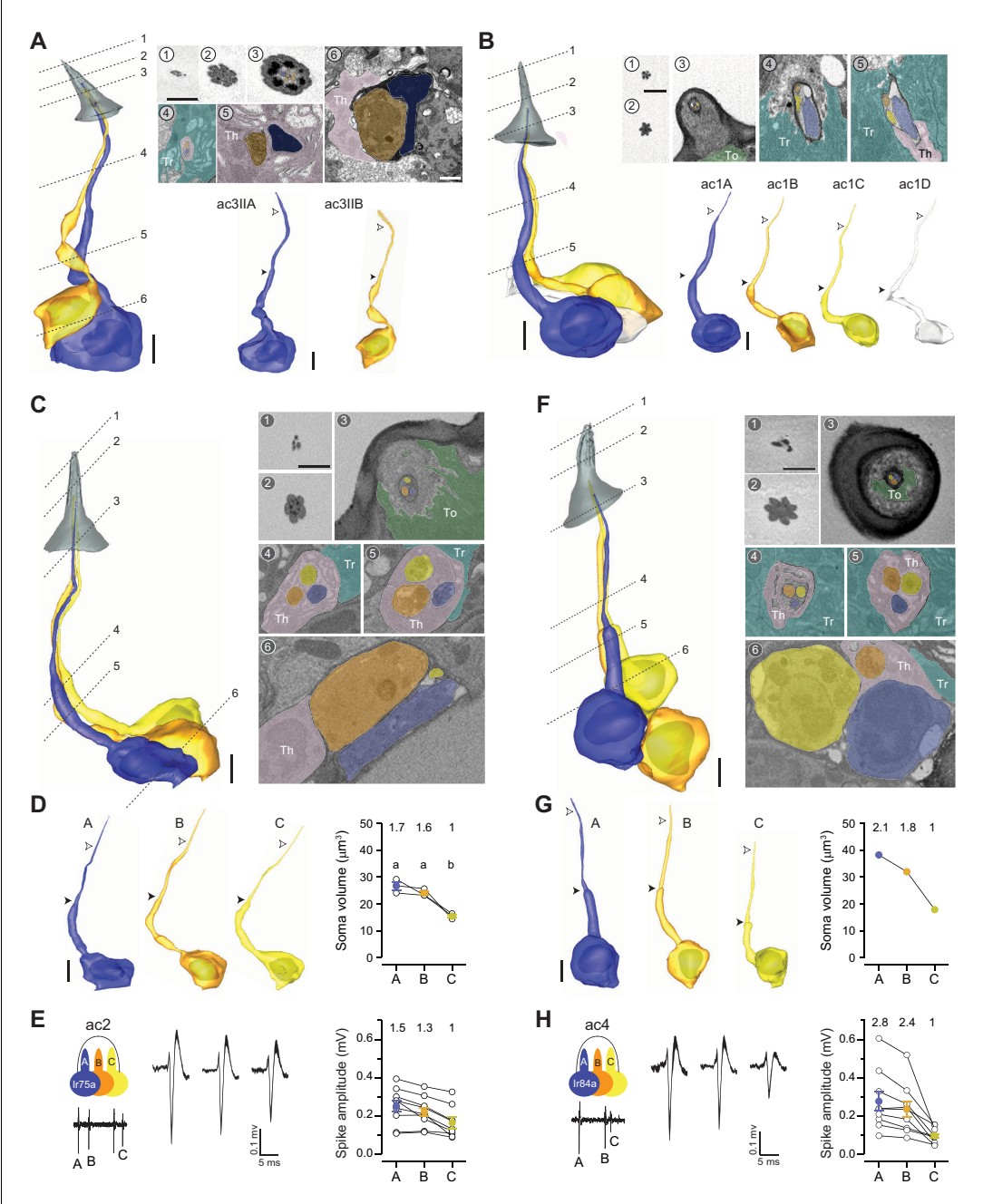

**Figure 4.** Coeloconic sensilla. (**A**) 3D model and SBEM images of a representative ac3 sensillum. The ac3A ORNs were engineered to express APEX2 (*Ir75c-GAL4>10XUAS-myc-APEX2-Orco*) and labeled with DAB (darker staining). ORNs are pseudocolored to indicate neuronal identity: ac3A (blue) and ac3B (orange). Dashed lines indicate positions of the corresponding SBEM images. Filled arrow heads: ciliary constriction; empty arrow heads: sensillum base position. Th: thecogen; Tr: trichogen; and To: tormogen. (**B**) A representative ac1 sensillum. The four ORNs are pseudocolored based on their relative soma size in descending order: blue, orange, yellow, and white. (**C**) A representative ac2 sensillum. The three ORNs are pseudocolored based on their relative soma size in descending order: blue, orange, and yellow. (**D**) 3D models for individual ac2 neurons as shown in (**C**) and their respective soma volumes. Lines connect data points from neurons housed in the same sensillum, mean± s.e.m. n=3. The soma volume ratio is indicated above the graph. Significant differences (p<0.05) are indicated by different letters; paired t-test for comparison between neighboring neurons. (**E**) Extracellular spike amplitudes of grouped ac2 ORNs. Left: Average spike waveforms are shown for individual neurons. Line width indicates s.e.m. Right: Spike amplitude comparison. Each data point represents the average spike amplitude of an ORN based on its spontaneous activity. Lines connect data points from neighboring neurons. Colored dots denote average spike amplitudes, mean± s.e.m. n=9. Spike amplitude ratio is indicated above the graph. (**F–H**) As in (**C–E**), except that images are shown for an ac4 sensillum. Scale bars: 2 µm for 3D models and 1 µm for SBEM images. The scale bar

*Figure 4 continued on next page*

*Figure 4 continued*

in the first image panel also pertains to other images unless indicated otherwise. DAB, diaminobenzidine; ORN, olfactory receptor neuron; SBEM, serial block-face scanning electron microscopy.

The online version of this article includes the following figure supplement(s) for figure 4:

**Figure supplement 1.** Atypical ac1 sensillum.

Compared to the ac3 sensillum, ac1's cuticular fingers have a sharper edge, giving rise to its star-shaped cross-section profile (*Figure 4B*). The somatic volume of ac1A is nearly twice as large (~23 $\mu m^3$) as ac1B (~12 $\mu m^3$), which is in turn larger than ac1C (~10 $\mu m^3$) and ac1D (~8 $\mu m^3$) (*Figure 4B*, *Figure 4—figure supplement 1A*, and *Table 2*). Interestingly, the rank order for inner dendritic volume is A>B>D>C, whereas for the outer dendritic volume it is A>D>B=C (*Figure 4—figure supplement 1B,C*). With respect to their dendritic morphology, all ac1 neurons have unbranched outer dendrites (*Table 2*). Of note, among the five groups of segmented ac1 neurons, one 'A' neuron exhibited a considerably enlarged inner dendrite (*Figure 4—figure supplement 1* and *Video 3*). The inner dendrite of this atypical neuron contained 24 mitochondria, as opposed to 4–6 for other ac1A neurons (see *Table 2—source data 1* for details). Although the functional significance of this inner dendritic enlargement remains unclear, it will be interesting to determine whether this atypical feature reflects an ongoing dynamic cellular event or marks a distinctive group of neurons in an ac1 sensillum subtype.

## ac2 and ac4 sensilla

We identified two types of coeloconic sensilla that house three ORNs, which likely correspond to the ac2 and ac4 sensilla (*Benton et al., 2009*). The ac2 ORNs express the Ir75a, Ir41a, and Ir75d receptors, while the ac4 neurons express the Ir84a, Ir75d, and Ir76a receptors, respectively (*Benton et al., 2009*; *Silbering et al., 2011*). We distinguished these two sensillum types based on their relative ORN soma size. Among the two sensillum populations, one compartmentalizes neurons whose soma sizes have ratios of 1.7:1.6: 1 (*Figure 4C,D*), close to the spike amplitude ratio for the ac2 neurons (*Figure 4E*). In comparison, the other sensillum type houses ORNs whose soma size ratio is 2.1:1.8:1 (*Figure 4F,G*), likely corresponding to ac4, which has a larger A-to-C spike ratio than ac2 (*Figure 4E,H*). Apart from relative ORN soma size, the two sensilla can be further differentiated by their external morphology. The cuticular fingers are rounder for ac2 than for ac4, such that the cross-section profile has an appearance of a flower for ac2, as opposed to a star for ac4 (*Figure 4C and F*). As with the ac1 neurons, the outer dendrites for ac2/ac4 ORNs are unbranched (*Table 2* and *Video 3*).

## Intermediate sensilla

The intermediate sensilla share similar cuticular morphology with trichoid sensilla (*Figure 1A,B*), but their length is shorter than that of trichoids (*Figure 1D* and *Table 1*). The intermediate sensilla also display the basal drum at the sensillum base which, although not as prominent as the trichoid counterpart, can be recognized in isosurface reconstructions (*Figure 1C*) or by the thick cuticular walls in SBEM sections (*Figure 5A*). Another distinctive feature for intermediate sensilla is the darkly stained lumen which contrasts with the light luminal staining in the basiconic and trichoid sensilla from the same SBEM data set (*Figure 5A*).

In agreement with an earlier TEM study (*Shanbhag et al., 1999*), we also identified two populations of intermediate sensilla that house either two or three neurons, likely corresponding to the ai2 and ai3 sensilla, respectively. The ai2 ORNs express the Or83c (ai2A) and Or23a (ai2B) receptors, while the ai3 neurons express the Or19a (ai3A), Or2a (ai3B), and Or43a receptors (ai3C) (*Dweck et al., 2013*; *Lin and Potter, 2015*; *Ronderos et al., 2014*). On average, ai2 is ~3 µm longer than ai3 (*Figure 1D* and *Table 1*). For the ai2 sensillum, the A neuron soma is approximately 50% larger than the neighboring B neuron, and the size differential is greater for the inner dendritic segment, and still greater for the outer dendritic segment (*Figure 5B-D* and *Table 2*). All segmented ai2 neurons exhibit branched outer dendrites (*Figure 5B*, *Figure 5—figure supplement 1*, and *Video 4*) with branches numbering from 5 to 11 for ai2A and 3 to 4 for ai2B (*Table 2*). Of note, striking morphological variability was observed for ai2A dendrites. While the majority of branches are

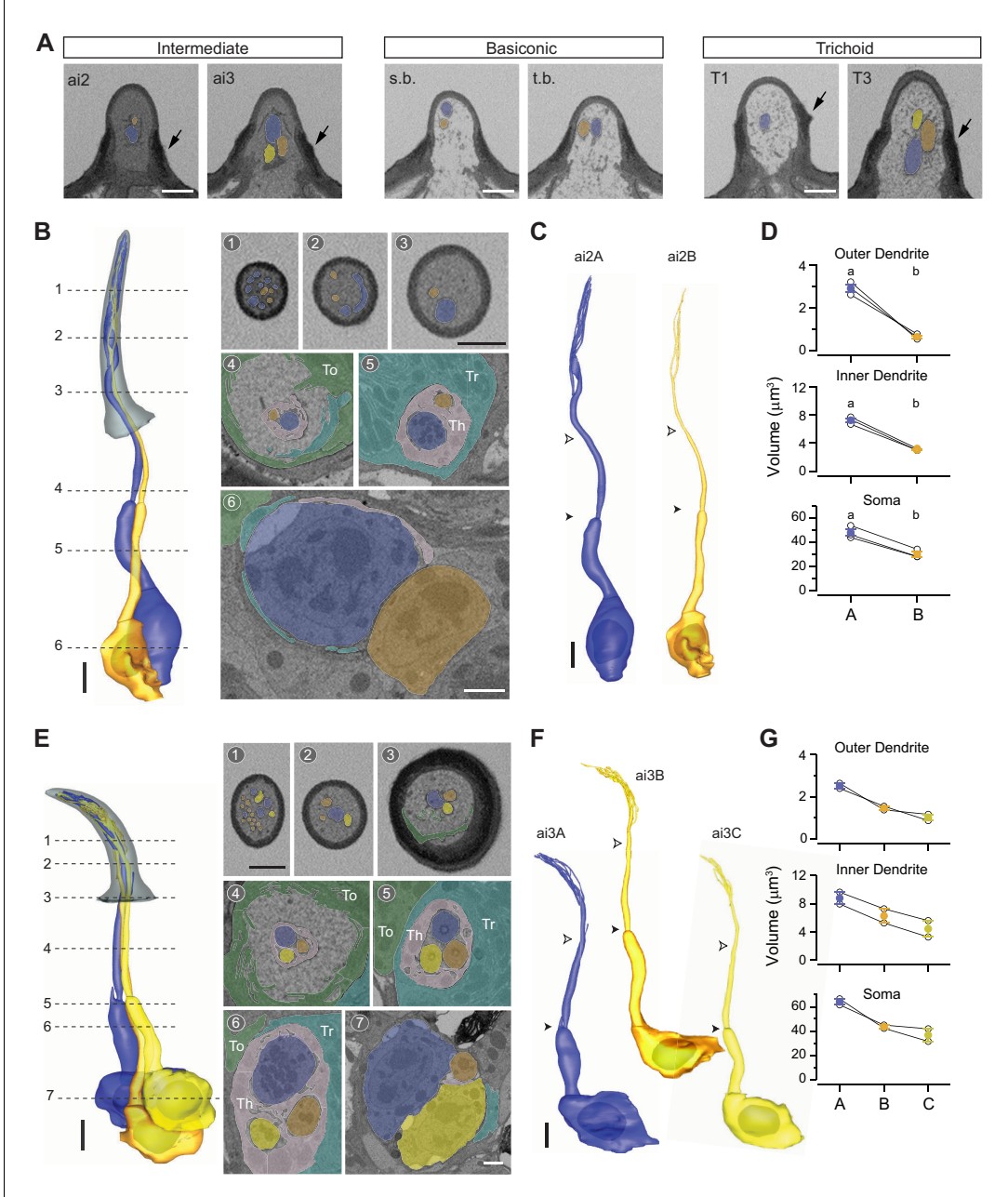

**Figure 5.** Intermediate sensilla. (**A**) SBEM images of representative intermediate, basiconic, and trichoid sensilla sampled from the same data set. Intermediate sensilla display higher luminal density than the other two sensillum types. Arrows: basal drum; s.b. small basiconic; and t.b. thin basiconic. (**B**) 3D model and SBEM images of a representative ai2 sensillum. ORNs are pseudocolored to indicate neuronal identity: ai2A (blue) and ai2B (orange). Dashed lines indicate positions of the corresponding SBEM images. Th: thecogen; Tr: trichogen; and To: tormogen. (**C**) 3D models for individual ai2 neurons as shown in (**B**). Filled arrow heads: ciliary constriction; empty arrow heads: sensillum base position. (**D**) Volumes of the soma, inner, and outer dendrites of compartmentalized ai2 neurons. Lines connect data points from neurons housed in the same sensillum, mean± s.e.m. n=3. Significant differences (p<0.05) are indicated by different letters; paired t-test for comparison between neighboring neurons. (**E–G**) As in (**B–D**), except that images are shown for a representative ai3 sensillum. ORNs are pseudocolored to indicate neuronal identity: ai3A (blue), ai3B (orange), and ai3C (yellow). n=2. Scale bars: 2 µm for 3D models and 1 µm for SBEM images. The scale bar in the first image panel also pertains to other images unless indicated otherwise. ORN, olfactory receptor neuron; SBEM, serial block-face scanning electron microscopy.

The online version of this article includes the following figure supplement(s) for figure 5:

**Figure supplement 1.** Diverse dendritic morphologies of ai2A ORNs.

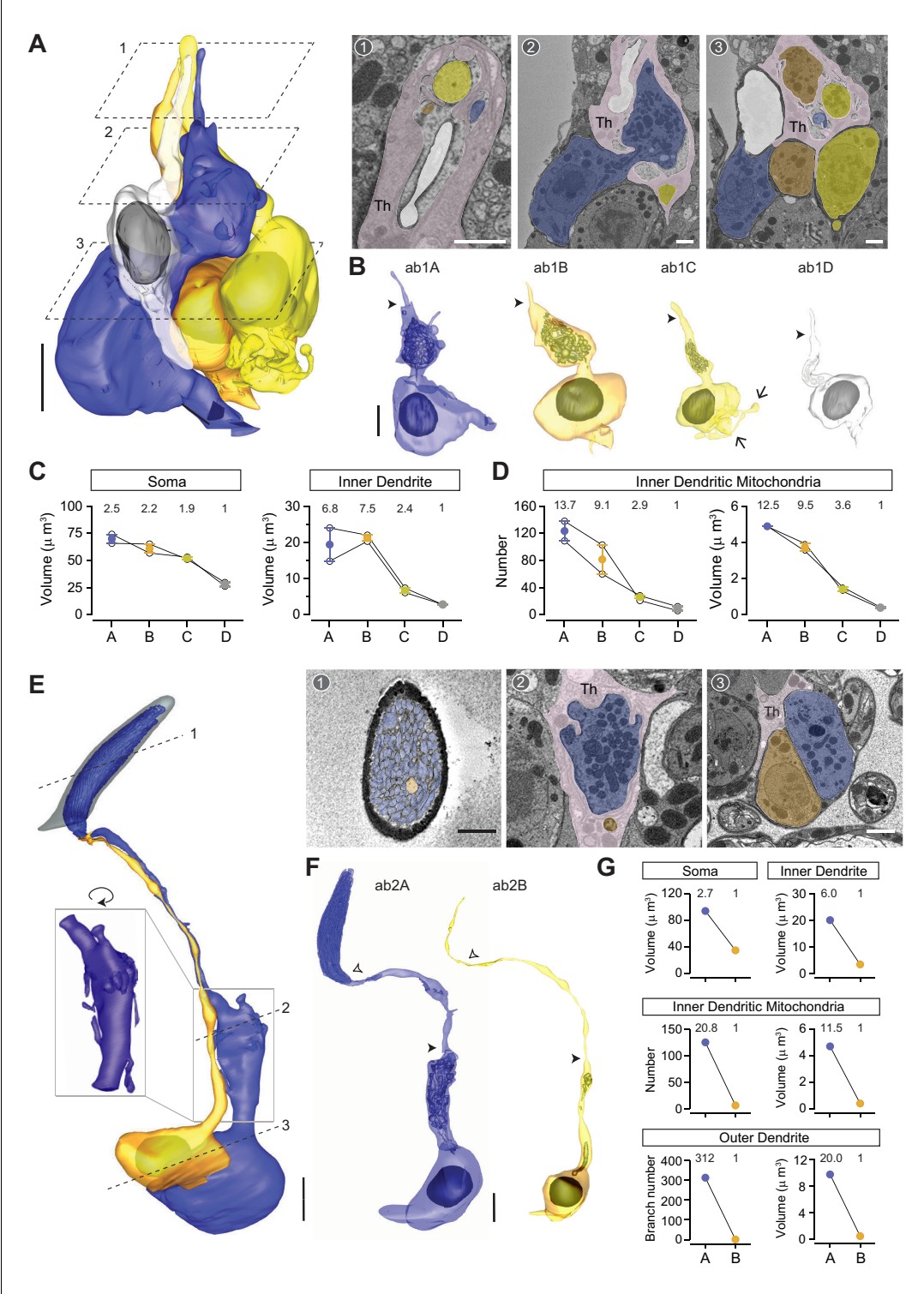

**Figure 6.** ab1 and ab2 large basiconic sensilla. (A) A partial 3D model and sample SBEM images of a representative ab1 sensillum. ORNs are pseudocolored to indicate neuronal identity: ab1A (blue), ab1B (orange), ab1C (yellow), and ab1D (white). Dashed planes indicate positions of the corresponding SBEM images. Among the auxiliary cells, only thecogen is indicated (Th) because the identity for trichogen or tormogen cannot be unambiguously determined. (B) 3D models for individual ab1 neurons as shown in (A). Filled arrow heads: ciliary constriction. Arrow: membranous protrusion. (C) Volumes of the soma, inner, and outer dendrites of compartmentalized ab1 neurons. (D) The number and total volume of mitochondria in the inner dendritic segments of ab1 neurons. Lines connect data points from neurons housed in the same sensillum, mean± s.e.m. n=2. The volume or number ratio is indicated above each graph. (E, F) As in (A, B), except that a complete 3D model of an ab2 sensillum is shown. Filled arrow heads:

*Figure 6 continued on next page*

*Figure 6 continued*

ciliary constriction; empty arrow heads: sensillum base position. Inset: magnified and rotated view, highlighting the inner dendritic membranous protrusions. (G) Morphometric comparison of ab2A and ab2B neurons. The volume or number ratio is indicated above each graph. Scale bars: 2 μm for 3D models and 1 μm for SBEM images. The scale bar in the first image panel also pertains to other images unless indicated otherwise. ORN, olfactory receptor neuron; SBEM, serial block-face scanning electron microscopy.

cylindrical with a gradual taper toward the tip, we found one that was flattened like a ribbon which spiraled upwards (*Figure 5—figure supplement 1A*). In another ai2A, the flattened region was observed in the unbranched segment of outer dendrite, which then gave rise to multiple cylindrical and some flattened sub-branches (*Figure 5—figure supplement 1B*). As for the ai3 sensilla, a similar size difference was observed for their grouped neurons: the 'A' neuron exhibits the largest soma, inner, and outer dendrites, and is followed by the neighboring ai3B and then ai3C neurons (*Figure 5E-G*). As with ai2, all analyzed ai3 neurons have branched outer dendrites (*Table 2* and *Video 4*). Finally, extracellular vacuoles were observed in the lumen among 21% and 56% of the analyzed ai2 and ai3 sensilla, respectively.

## Large basiconic sensilla ab1 sensillum

Characteristic of this basiconic subtype is the large, club-shaped cuticular apparatus which has a rounded tip and a slight narrowing at the base (*Figure 1C*). Large basiconic sensilla range from 10 to 13 μm in length (*Figure 1D* and *Table 1*). Among large basiconics, ab1 is unique because it houses not two but four neurons (*Figure 6A,B*), which express the Or42b (ab1A), Or92a (ab1B), Gr21a/Gr63a (ab1C), and Or10 receptors (ab1D) (*Couto et al., 2005*; *de Bruyne et al., 2001*; *Hallem et al., 2004*; *Jones et al., 2007*; *Kwon et al., 2007*). Compartmentalized ab1 neurons have distinctive sizes, with the soma volumes in the ratio of 2.5:2.2:1.9:1, following the rank order A≥B>C>D (*Figure 6C*). Interestingly, ab1C, which senses $CO_2$ (*de Bruyne et al., 2001*; *Jones et al., 2007*; *Kwon et al., 2007*), exhibits multiple membranous protrusions that resemble neuronal processes extending from the basolateral side of the soma (*Figure 6B*, arrows). Because of this unique feature, ab1C soma has the second largest surface area among the four grouped neurons (A>C>B>D), despite its relatively small soma volume (*Table 2*).

More interestingly still, both ab1A and ab1B display markedly enlarged inner dendritic segments such that the size differential of ab1 neurons is even greater for this region—with a volumetric ratio of 6.8:7.5:2.4:1 (*Figure 6B,C*). As with the atypical ac1A (*Figure 4—figure supplement 1*), these inner dendritic enlargements contain numerous mitochondria (*Figure 6A,B*), with an average number of 124 for ab1A and 82 for ab1B (*Figure 6D* and *Table 2*). In comparison, the neighboring small-spike 'C' and 'D' neurons contain an average of 26 and 9 mitochondria, respectively (*Figure 6D*, *Table 2* and *Video 5*). The disparity in mitochondrial number is also reflected in total mitochondrial volume (*Figure 6D*). We note that in vertebrate ORNs, mitochondria play a direct role in regulating cytosolic $Ca^{2+}$ response profiles and thereby ensure a broad dynamic range for the neurons' spike responses (*Fluegge et al., 2012*). Although it is unclear whether mitochondria play a similar role in insect olfactory signaling, a recent study shows that odor-induced $Ca^{2+}$ signals in *Drosophila* ORNs are shaped by mitochondria (*Lucke et al., 2020*). Therefore, it will be interesting to investigate the functional significance of this striking mitochondrial disparity between grouped ORNs in future research.

## ab2 and ab3 sensilla

Among the two large basiconic sensilla that house two neurons, we distinguished ab2 from ab3 by its lack of DAB staining in the Or22a data set, in which ab3A was genetically labeled by APEX2. Apart from the Or22a data set, an ab2 sensillum was identified in the Or7a data set on the basis of its proximity to the labeled ab4 sensillum, because ab3 is not found in the same topographical region as ab4 (*de Bruyne et al., 2001*). The ab2 and ab3 receptors are Or59b (ab2A), Or85a (ab2B), Or22a (ab3A), and Or85b (ab3B) (*Dobritsa et al., 2003*; *Hallem et al., 2004*). On average, the cuticular volume of ab2 (~32 $\mu m^3$) is smaller than that of the genetically identified ab3 (~39 $\mu m^3$) (*Table 1*). Among the four ab2 sensilla found in our data sets, we were able to segment a pair of ab2 ORNs in their entirety (*Figure 6E,F* and *Video 5*). In agreement with ab2's large A/B spike amplitude ratio (*Zhang et al., 2019*), the soma size differential is notably high for the two neurons (2.7:1)

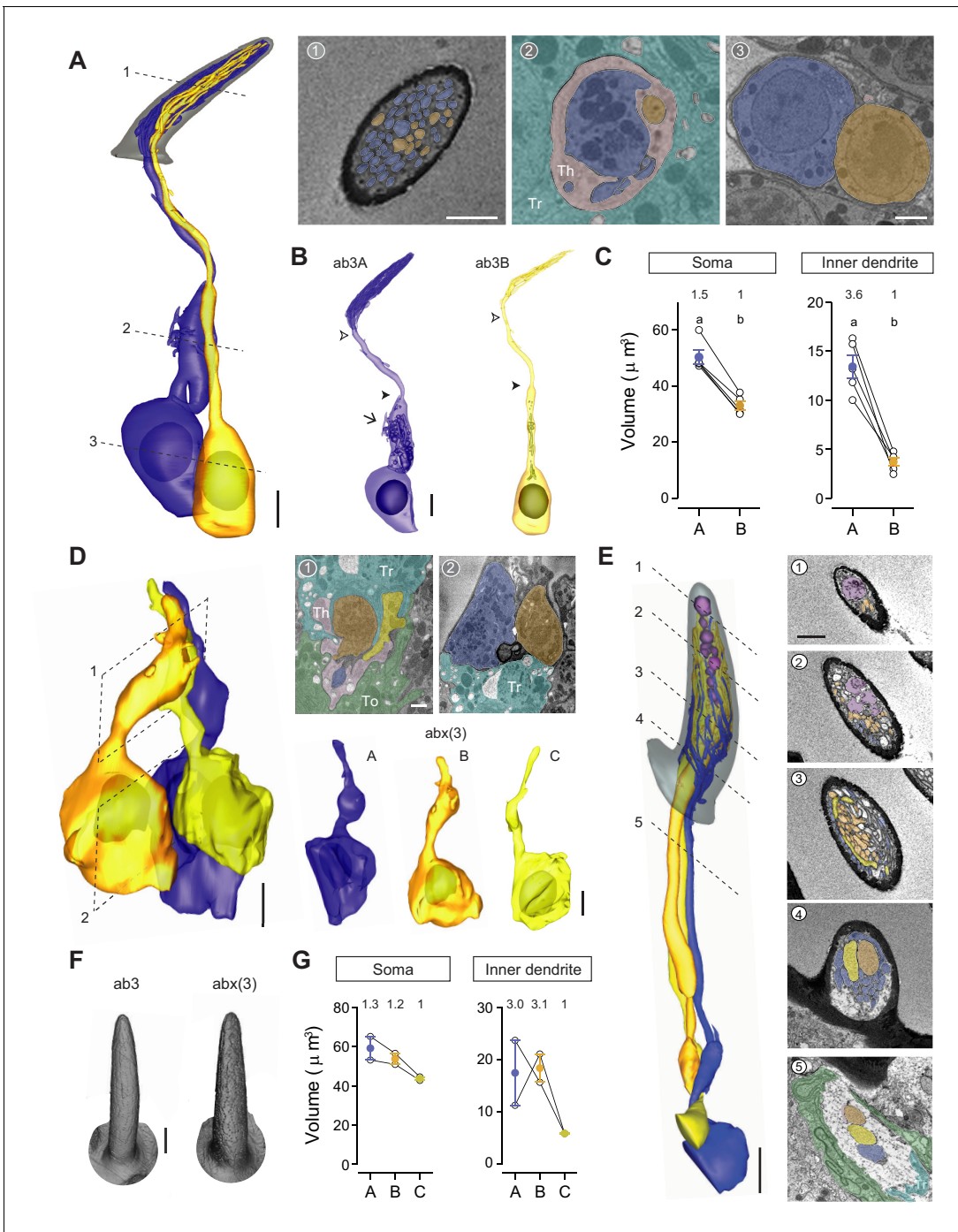

**Figure 7.** ab3 and abx(3) large basiconic sensilla. (**A**) 3D model and sample SBEM images of an ab3 sensillum. ORNs are pseudocolored to indicate neuronal identity: ab3A (blue) and ab3B (orange). Dashed lines indicate positions of the corresponding SBEM images. Th: thecogen; Tr: trichogen; and To: tormogen. (**B**) 3D models for individual ab3 neurons as shown in (**A**). Filled arrow head: ciliary constriction; empty arrow head: sensillum base position. Arrow: membranous protrusions. (**C**) Volumes of the soma and inner dendrites of grouped ab3 neurons. Lines connect data points from individual neurons housed in the same sensillum, mean± s.e.m. n=5. The volume ratio is indicated above each graph. Significant differences (p<0.05) are indicated by different letters; paired t-test for comparison between neighboring neurons. (**D, E**) Partial 3D models and sample SBEM images of different abx(3) sensilla. ORNs are pseudocolored to indicate neuronal identity: abx(3)A (blue), abx(3)B (orange), and abx(3)C (yellow). Dashed planes or lines indicate positions of the corresponding SBEM images. (**D**) Partial soma and inner dendritic models, and (**E**) partial outer dendritic models. The identity for some dendritic branches cannot be unambiguously determined and are thus not pseudocolored. (**F**) Isosurface images of ab3 and abx(3) sensilla. (**G**) Volumes of the soma and inner dendrites of grouped abx(3) neurons. Lines connect data points from neurons housed in the same sensillum, mean± s.e.m. n=2. The volume ratio is indicated above each graph. Scale bars: 2 μm for 3D models and 1 μm for SBEM images. The scale

*Figure 7 continued on next page*

*Figure 7 continued*

bar in the first image panel also pertains to other images from the same sensillum unless indicated otherwise. ORN, olfactory receptor neuron; SBEM, serial block-face scanning electron microscopy.

(*Figure 6G*, top left panel). Of note, ab2A but not ab2B displays a bulbous inner dendritic enlargement and numerous long membranous protrusions (*Figure 6E* inset), which further widens the size disparity between the two neurons for this region (6:1) (*Figure 6G*, top right panel). As with the large-spike ab1 neurons, the ab2A inner dendritic enlargement contains over 100 mitochondria, markedly more numerous and with a greater total volume than ab2B (*Figure 6G*, middle panels, and *Table 2*). Moreover, an even greater disparity was observed in the outer dendritic segments: ab2A exhibits 312 dendritic branches while ab2B has only one (*Figure 6G*, bottom panels).

With respect to the ab3 large basiconic sensilla, four pairs of ORNs were identified and partially segmented from the Or22a-labeled data set in which the ab3A neurons were marked genetically. Additionally, we segmented a pair of ab3 neurons in their entirety from another data set (*Figure 7A, B* and *Video 5*), in which the ab3 sensillum was identified on the basis of its characteristic morphometric features (*Table 1*). Morphometric analysis of the soma and inner dendrite revealed a marked size difference between ab3A and ab3B (*Figure 7C*), as described in our previous work (*Zhang et al., 2019*). As with ab2A, the inner dendrite of ab3A also contains numerous mitochondria and displays several long membranous protrusions (*Figure 7A,B*). Similar to the other large basiconic neurons, the inner dendritic size disparity for grouped ab3 neurons (3.6:1) is greater than the somatic size difference (1.5:1) (*Figure 7C*). The outer dendritic disparity for ab3 neurons is less striking when compared to ab2 ORNs; the completely segmented ab3A and ab3B exhibit 83 and 19 dendritic branches, respectively (*Table 2*).

## Novel large basiconic sensillum housing three ORNs

In addition to the three known large basiconic sensilla (ab1–ab3), we identified a novel type that houses three ORNs (*Figure 7D,E*). As shown in the isosurface images and 3D models, this novel sensillum, which we termed abx(3), and the ab3 sensillum are indistinguishable based on their exterior morphological and morphometric features (*Figure 7F* and *Table 1*). Although we were unable to segment the abx(3) ORNs in their entirety, partial 3D reconstruction of the soma/inner dendrite or the outer dendrites was performed for different groups of abx(3) neurons (*Figure 7D,E* and *Video 5*). Compared to other compartmentalized large basiconic ORNs, the soma size disparity among the abx(3) neurons is less pronounced, with a ratio of 1.3:1.2:1 (*Figure 7G*). Nevertheless, a marked inner dendritic size difference was observed (3.0:3.1:1) (*Figure 7G*), because both A and B neurons but not the C neuron exhibit mitochondria-enriched enlargement (*Figure 7D*). We note that the size similarity between abx(3)A and abx(3)B neurons is reminiscent of that between ab1A and ab1B (*Figure 6C*), raising the possibility that abx(3) represents a subset of ab1 sensilla lacking the ab1D neuron. If so, it will be interesting to determine whether abx(3)C also corresponds to ab1C, given the lack of membranous protrusions in abx(3)C (*Figure 7D*). We note it is also possible that abx(3) represents an ab3 subset, or instead houses three uncharacterized orphan ORNs whose receptors have not yet been reported.

With respect to the outer dendrites of abx(3) neurons, oblique section angles prevented the reliable assignment of neuronal identity to every dendritic branch (*Figure 7E*). Nevertheless, the segmented portion was sufficient to reveal some interesting features. Notably, the outer dendrite of abx(3)A started branching extensively near the sensillum base, well before branching was observed in the neighboring B or C neurons (*Figure 7E*). Finally, extracellular vacuoles were found in the distal lumen (*Figure 7E*), in contrast to the vacuoles in the trichoid or intermediate sensilla, which were predominately located near the sensillum base. Future research may be directed to determine whether the location of vacuoles influences their function. We also note that more than 90% of the examined large basiconic sensilla exhibit vacuoles, making them the most prominent sensillum type to display these extracellular membranous structures.

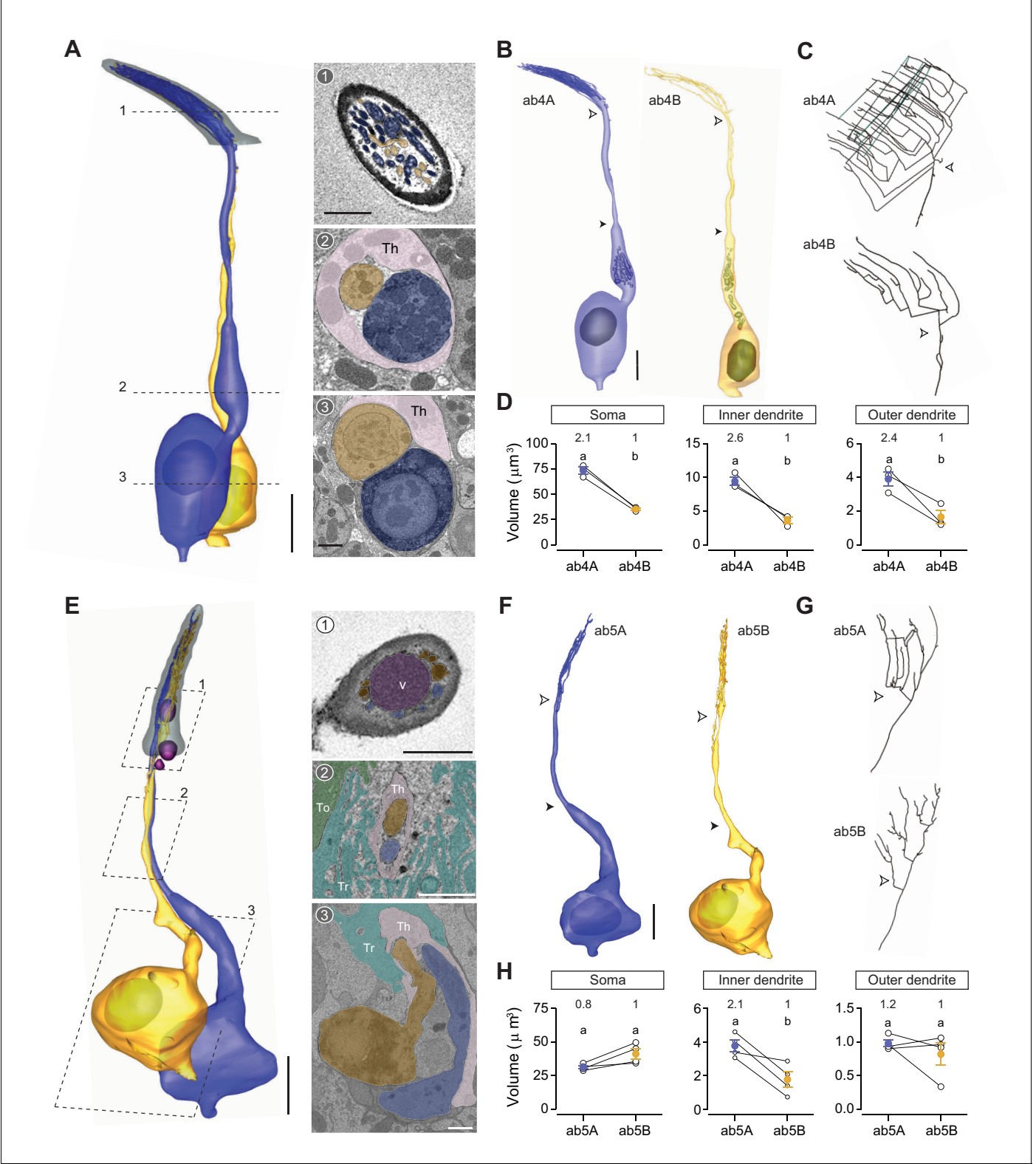

**Figure 8.** ab4 thin basiconic sensillum and ab5 small basiconic sensillum. (**A**) 3D model and sample SBEM images of a representative ab4 sensillum. The ab4A ORNs were engineered to express APEX2 (*Or7a-GAL4>10XUAS-myc-APEX2-Orco*) and labeled with DAB (darker staining). ORNs are pseudocolored to indicate neuronal identity: ab4A (blue) and ab4B (orange). Dashed lines indicate positions of the corresponding SBEM images on the right. (**B**) 3D models for individual ab4 neurons as shown in (**A**). Filled arrow heads: ciliary constriction; empty arrow heads: sensillum base position. (**C**) 2D projections of ab4A or ab4B outer dendritic branches. (**D**) Morphometric comparison of ab4A and ab4B neurons. Lines connect data points from

*Figure 8 continued on next page*

*Figure 8 continued*

neurons housed in the same sensillum, mean± s.e.m. n=3. The volume ratio is indicated above each graph. (E–H) As in (A–D), except that ab5 is featured. The ab5B ORNs were engineered to express APEX2 (*Or47a-GAL4>10XUAS-myc-APEX2-Orco*) and labeled with DAB (darker staining). Four pairs of ab5 neurons were analyzed for morphometrics. Significant differences (p<0.05) are indicated by different letters; paired t-test for comparison between neighboring neurons. Th: thecogen; Tr: trichogen; and To: tormogen. Scale bars: 2 μm for 3D models and 1 μm for SBEM images. The scale bar in the first image panel also pertains to other images unless indicated otherwise. DAB, diaminobenzidine; ORN, olfactory receptor neuron; SBEM, serial block-face scanning electron microscopy.

The online version of this article includes the following figure supplement(s) for figure 8:

**Figure supplement 1.** Novel small basiconic sensillum with one ORN.

## Thin and small basiconic sensilla ab4 sensillum

All seven molecularly defined thin and small basiconic sensillum types (ab4–ab10) house two ORNs (*Couto et al., 2005*). We therefore focused our analysis on the genetically labeled ab4 and ab5 sensilla, which express Or7a (ab4A), Or56a (ab4B), Or82a (ab5A), and Or47a (ab5B) as odor-tuning receptors (*Hallem et al., 2004*; *Stensmyr et al., 2012*). Matching the description of a thin basiconic (*Shanbhag et al., 1999*), the ab4 sensillum is similar to the large basiconics in length (~11 μm) but with a more slender profile (*Figure 1C,D* and *Table 1*). Three pairs of ab4 neurons, including their elaborate dendritic branches, were completely segmented (*Figure 8A-C* and *Video 6*). Compartmentalized ab4 ORNs also have distinctive sizes, with the soma volume in the ratio of 2.1:1 (*Figure 8D*). Although the inner dendrite of ab4A is similarly enlarged and enriched with mitochondria (*Figure 8B*), the enlargement is less pronounced: ab4A's inner dendritic volume is ~9 $\mu m^3$ as opposed to ~20 $\mu m^3$ for ab1A or ab2A (*Table 2*). As such, the inner dendritic volume ratio for ab4A and ab4B is 2.6:1, which is close to the somatic size differential (*Figure 8D*). A similar volume difference was also observed in the outer dendrites (2.4:1) (*Figure 8D*).

In addition to morphometric analysis, we examined the branching pattern of ab4 outer dendrites and found it to be strikingly complex, contrary to the simple paintbrush model proposed on the basis of TEM longitudinal-section images of a sensillum, whereby all basiconic dendritic branches appear to share a common diverging point (*Shanbhag et al., 1999*). Instead in ab4A, we observed multiple branching points. A primary branch can bifurcate several times, giving rise to numerous secondary branches that in turn can give rise to tertiary, quaternary, or even higher-order branches. Moreover, some high-order branches can converge with others to form lateral connections (*Figure 8C*, top panel). In other instances, a branch can diverge into two sub-branches which then merge back into one (*Video 7*). In comparison to ab4A, the dendritic branching pattern of ab4B is less elaborate, with a smaller number of branches and a lower degree of branching (*Figure 8C*, bottom panel). Despite their varying complexity, the ab4A and ab4B outer dendrites start branching at similar positions near the sensillum base (*Figure 8A-C*).

## ab5 sensillum

Among antennal olfactory sensilla, ab5 is unique because its compartmentalized ORNs have similar extracellular spike amplitudes and morphometric features (*Zhang et al., 2019*). The average length of an ab5 sensillum is ~9 μm (*Figure 1C,D* and *Table 1*), matching the description of a small basiconic sensillum (*Shanbhag et al., 1999*). We segmented four pairs of ab5 neurons in their entirety (*Figure 8E-G* and *Video 6*). As described in our previous study (*Zhang et al., 2019*), grouped ab5 ORNs have similar soma volumes, but the inner dendrite of ab5A is larger than that of ab5B because of the mitochondria-rich enlargement (*Figure 8H* and *Table 2*). Nevertheless, no

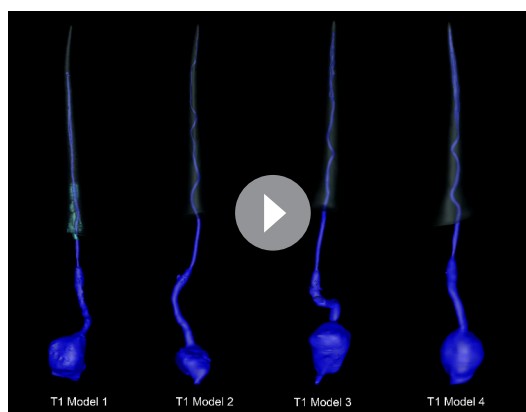

**Video 1.** 3D models of T1 olfactory receptor neurons. Color codes are as indicated in *Figure 2*.
https://elifesciences.org/articles/69896#video1

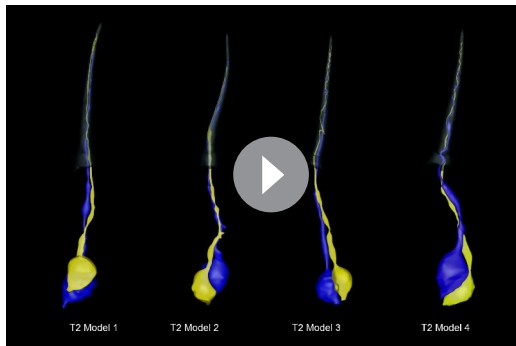

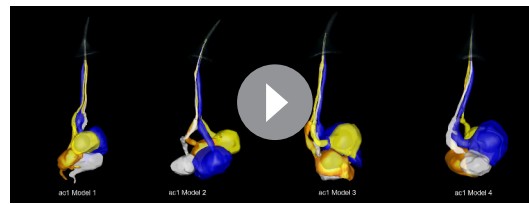

**Video 3.** 3D models of ac1–ac4 olfactory receptor neurons. Color codes are as indicated in *Figure 4*. https://elifesciences.org/articles/69896#video3

**Video 2.** 3D models of T2 and T3 olfactory receptor neurons. Color codes are as indicated in *Figure 3*. https://elifesciences.org/articles/69896#video2

significant difference in the outer dendritic volume was observed between paired ab5 neurons (*Figure 8H*). With respect to the ab5 outer dendritic branching pattern, it is similar to ab4 neurons by exhibiting multiple diverging points and high degrees of branching. The outer dendrites of grouped ab5 neurons also begin branching at similar positions near the sensillum base (*Figure 8G*). Finally, we note that approximately 70% of the examined thin and small basiconic sensilla exhibit extracellular vacuoles in the sensillum lumen (*Figure 8E*), suggesting that this structure is also common to this sensillum class.

## Novel small basiconic sensillum housing one ORN

Although most small basiconics house two ORNs, we unexpectedly identified a novel type that houses only one ORN (*Figure 8—figure supplement 1* and *Video 6*). We found three such sensilla, which we termed abx(1), in multiple SBEM data sets. As shown in the isosurface images and sensillum 3D models, abx(1) is similar to ab5 in length (~9 μm), characteristic of small basiconics (*Shanbhag et al., 1999*) but with a wider cuticular profile (*Figure 8—figure supplement 1A* and *Table 1*). Based on one complete and two partial 3D models, the neuron exhibits a slight inner dendritic enlargement and a small number of dendritic branches (*Figure 8—figure supplement 1B* and *Table 2*). In future research, it will be interesting to determine the receptor, ligands, and functional role for the singly housed abx(1) neuron.

## Empty sensilla

Among the 541 antennal sensilla sampled in this study, we were surprised to find two hairs that did not contain any ORNs. Such 'empty sensilla' have never been reported before. One is a trichoid sensillum identified from an Or47b-labeled data set (*Figure 9A*). Based on its cuticular morphometric features, the empty trichoid—named T(0)—likely belonged to the T1 subtype (*Table 1* and *Figure 1D*). Devoid of any neurons, the empty trichoid also did not contain any intact auxiliary cells; only remnants of what appeared to be tormogen processes were observed near the sensillum base (*Figure 9A*). The second empty sensillum is a small basiconic found in the Or7a-labeled data set (designated as abx(0), *Figure 9B* and *Table 1*). Interestingly, and unlike the empty trichoid, all three auxiliary cells were present and morphologically indistinguishable from those found in adjacent sensilla containing ORNs (*Figure 9C*). While the biological significance of empty sensilla remains to be determined, it is possible that they represent a small fraction of antennal olfactory hairs whose

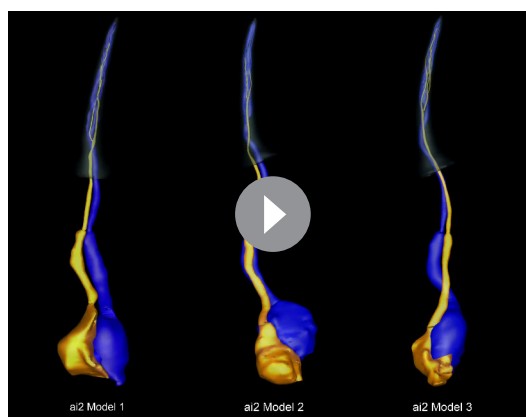

**Video 4.** 3D models of ai2 and ai3 olfactory receptor neurons. Color codes are as indicated in *Figure 5*. https://elifesciences.org/articles/69896#video4

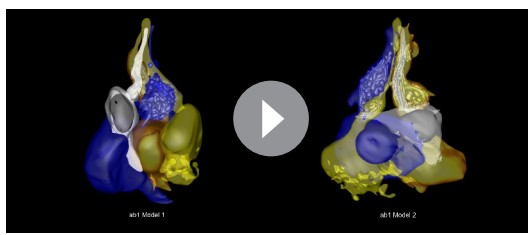

**Video 5.** 3D models of ab1–ab3 and abx(3) large basiconic neurons. Color codes are as indicated in *Figures 6* and *7*.
https://elifesciences.org/articles/69896#video5

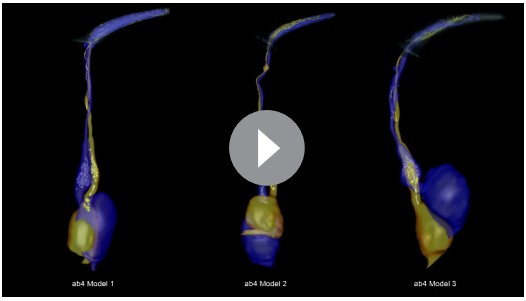

**Video 6.** 3D models of ab4, ab5, and abx(1) small basiconic neurons. Color codes are as indicated in *Figure 8*.
https://elifesciences.org/articles/69896#video6

neurons have undergone apoptosis in the process of neuronal turnover (*Fernández-Hernández and Hu, 2020*).

## Olfactory auxiliary cells

In addition to the sensilla and ORNs described in previous sections, we segmented two groups of auxiliary cells and reconstructed their 3D models. Auxiliary cells support olfactory function by insulating ORN soma and inner dendrites, maintaining transepithelial potential, expressing ion transporters, and secreting odorant-binding proteins and odorant degrading enzymes into the sensillum lymph (*Kaissling, 1986*; *Larter et al., 2016*; *Leal, 2013*; *Menuz et al., 2014*; *Shanbhag et al., 2001*; *Shanbhag et al., 2000*; *Sun et al., 2018*). We note that the ultrastructure of these cells has been described in detail through high-quality TEM images (*Shanbhag et al., 2000*). Therefore, our objective was to illustrate their positions in 3D space in relation to other auxiliary cells, their associated ORNs, and the sensillum apparatus. Most olfactory sensilla contain three auxiliary cells, except for the coeloconic class which has four (*Shanbhag et al., 2000*). In the following sections, we featured the auxiliary cells for ab4 and ac3 to represent these two possible cellular arrangements.

### Auxiliary trio

The three auxiliary cell types are the thecogen, trichogen, and tormogen cells (*Shanbhag et al., 2000*). Among them, the thecogen is situated most deeply beneath the antennal surface, and forms a sleeve enveloping parts of the ORN soma, inner dendrite, and proximal portion of the outer dendrite (*Figure 10A-C* and *Video 8*). The plasma membrane of thecogen cell is typically smooth but can form microvilli or microlamellae in regions where the auxiliary cell ensheathes the inner dendrites (e.g., *Figure 10C*; *Figures 3B-4* and *Figures 6A-1*). The nucleus of the auxiliary cell is at a similar level as the ORN soma such that thecogen cells are often found adjacent to the cell body of neurons (e.g., *Figure 10A-C*; *Figures 2A-9*, *Figures 3A-5 and B-5*).

The distal region of the thecogen cell—where it envelops the inner dendrites—is in turn enwrapped by the trichogen cell. The apical surface of the trichogen cell forms extensive, elaborate microvilli and microlamellae that border the sensillum-lymph cavity (*Figure 10A,B and D*). Also characteristic of the thecogen cell are its abundant mitochondria and numerous tubular membrane invaginations (*Figure 10A,B and D*). The nucleus of the trichogen cell is located slightly distal to that of the thecogen cell at the level of proximal inner dendrite (*Figure 10A* and *Video 8*).

Moving toward the antennal surface, a small portion of the apical thecogen cell is further

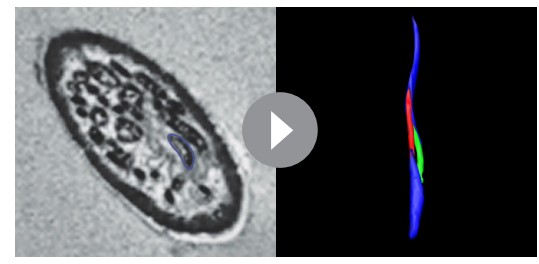

**Video 7.** SBEM images and 3D model of a subset of ab4A outer dendrites, demonstrating diverging and converging branches. The images were from Or7a-labeled data set. SBEM, serial block-face scanning electron microscopy.
https://elifesciences.org/articles/69896#video7

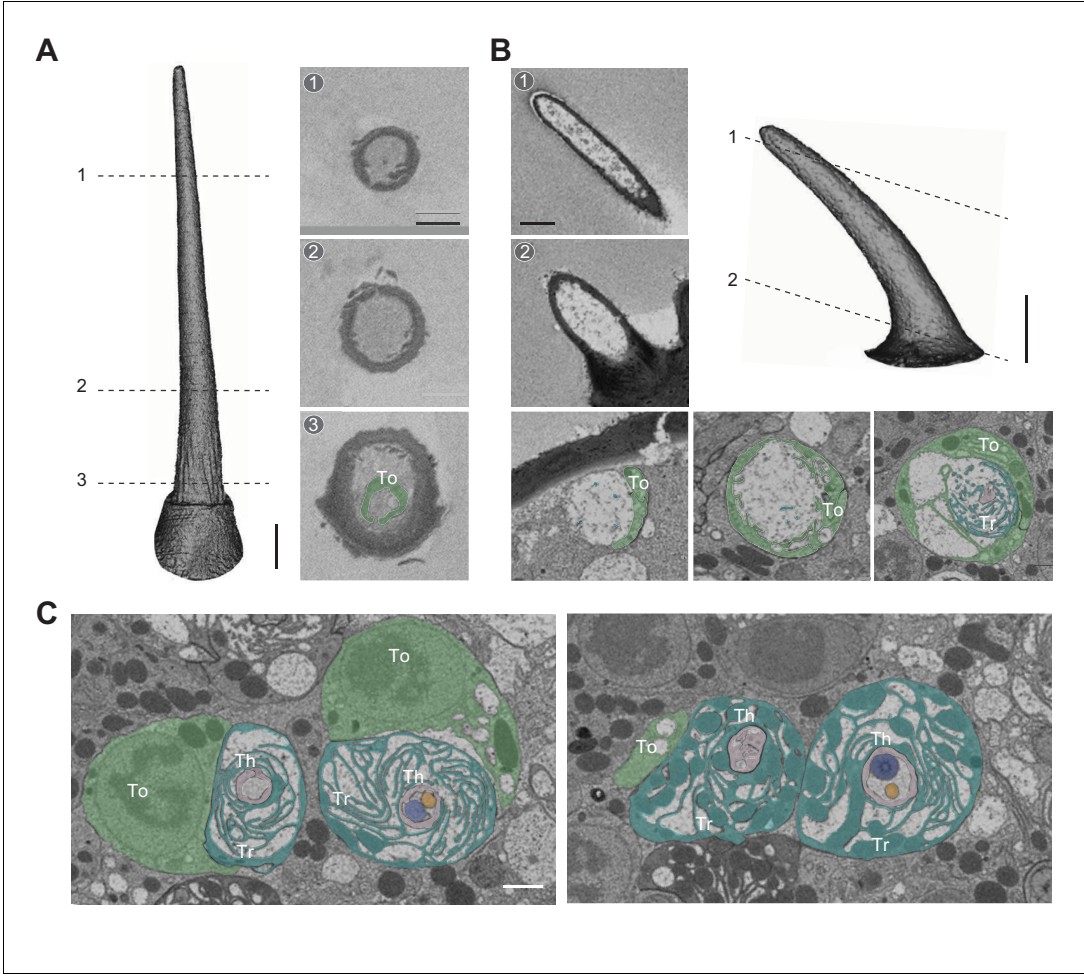

**Figure 9.** Empty sensillum. Isosurface and SBEM images of an empty T1 (**A**) or an empty small basiconic sensillum (**B**). Dashed lines indicate positions of the corresponding SBEM images. (**C**) SBEM images showing the three auxiliary cells associated with the empty small basiconic as shown in (**B**) or with a neighboring small basiconic containing two neurons. ORNs are pseudocolored in blue (A neuron) or orange (B neuron). The left image is closer to the antennal surface than the right image. Th: thecogen; Tr: trichogen; and To: tormogen. Scale bars: 2 μm for isosurface models and 1 μm for SBEM images. The scale bar in the first image panel also pertains to other images unless indicated otherwise. ORN, olfactory receptor neuron; SBEM, serial block-face scanning electron microscopy.

enveloped by the tormogen cell, which occupies the most distal and peripheral position of the auxiliary trio. Similar to the trichogen cell, the apical surface of the tormogen cell forms microvilli and microlamellae bordering the sensillum lymph cavity (*Figure 10A,B and E*). However, the tormogen processes are less numerous and elaborate. In turn, these processes surround the subcuticular segment of outer dendrites and terminate more distally near the sensillum base or occasionally into the sensillum lumen (e.g., *Figure 10A,B*; *Figures 2B* and *5E*). Moreover, in lieu of tubular invaginations, the tormogen cell displays an abundance of large, interconnected vesicles which open into the extracellular space (*Figure 10E*). The tormogen nucleus is situated at the level of dendritic ciliary constriction. The basal region of the tormogen cell is distinctive in that it extends a long, stalk-like protrusion that terminates below the level of ORN soma (*Figure 10E* and *Video 8*).

## Auxiliary quartet
Focusing on ac3, we segmented its four auxiliary cells, which have been reported to include one thecogen, one trichogen, and two tormogen cells (*Shanbhag et al., 2000*). Similar to the cellular arrangement described above, the coeloconic thecogen is the innermost cell located at the bottom

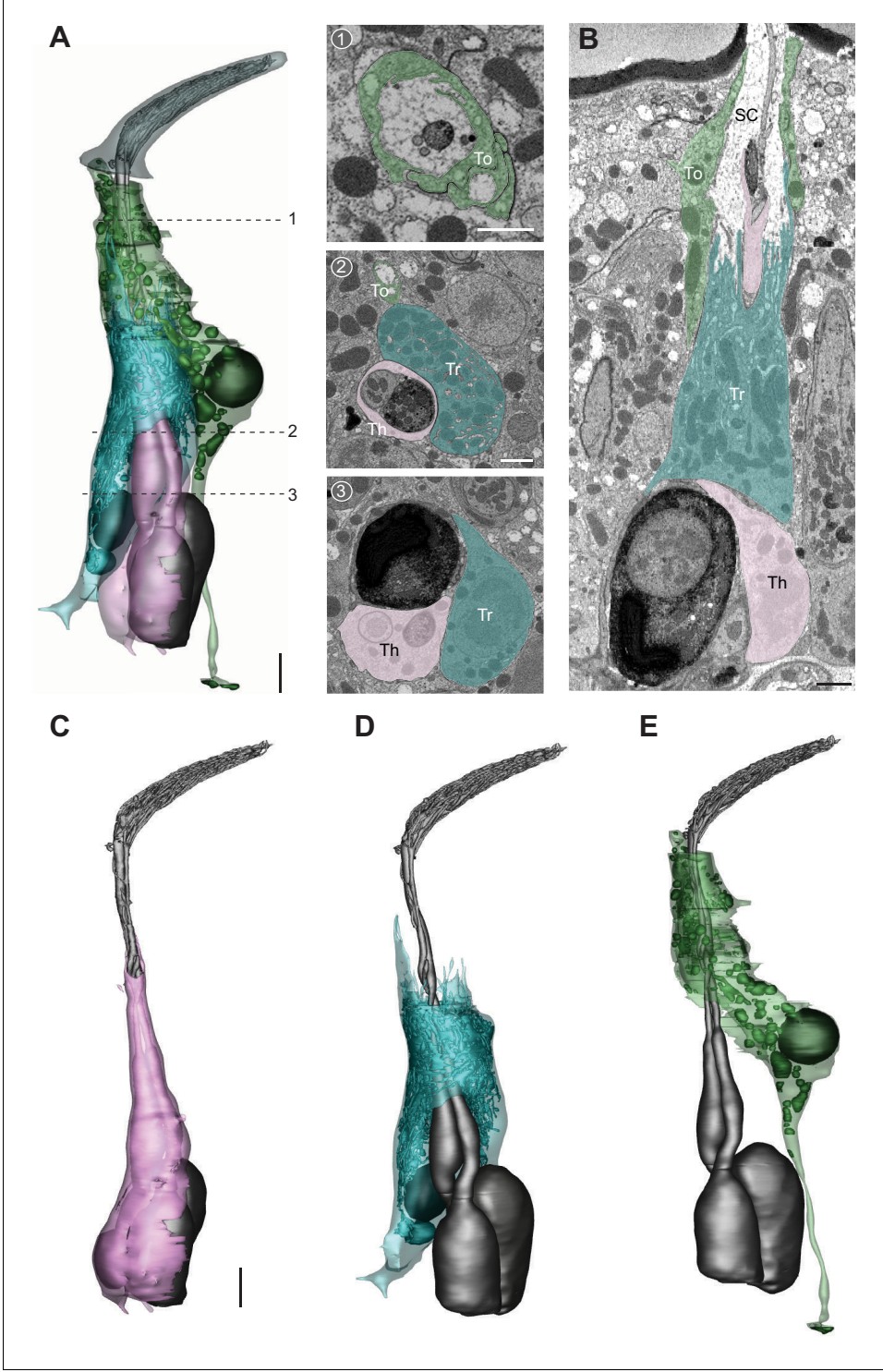

**Figure 10.** The three auxiliary cells for a basiconic sensillum. (**A**) 3D model and SBEM images of a representative ab4 sensillum, highlighting its three auxiliary cells. Images were taken from the Or7a-labeled data set. Cells are pseudocolored to indicate identities: ORNs (gray), thecogen cell (pink), trichogen cell (turquoise), and tormogen cell (green). Dashed lines indicate positions of the corresponding SBEM images. (**B**) IMOD software-rendered SBEM image of a longitudinal sensillum section. Th: thecogen; Tr: trichogen; To: tormogen; and SC: sensillum lymph cavity. (**C–E**) 3D models of individual auxiliary cells as shown in (**A**): thecogen (**C**), trichogen (**D**), and tormogen (**E**). Scale bars: 2 µm for 3D models and 1 µm for SBEM images. The scale bar in the first image panel

*Figure 10 continued*

also pertains to other images unless indicated otherwise. ORN, olfactory receptor neuron; SBEM, serial block-face scanning electron microscopy.

of the quartet, ensheathing parts of ORN soma, inner dendrite, and the proximal portion of outer dendrite (*Figure 11A-C* and *Video 8*). The apical thecogen region is likewise enveloped by trichogen processes, whose elaborate microvilli and microlamellae surround the proximal outer dendrites. Compared to the basiconic trichogen cell, the invaginations of coeloconic trichogen appear less tubular but are rather vesicle-like (*Figure 11D*). In addition, the coeloconic trichogen nucleus is located more distally, at or above the level of ciliary constriction. Interestingly, we identified two trichogen cells instead of one based on their similar morphological features, including elaborate microlamellae, the position of their apical surface, and the lack of tormogen's stalk-like protrusion (*Figure 11B,D*). This observation indicates that the coeloconic auxiliary quartet is in fact composed of one thecogen, one tormogen, and two trichogen cells. Finally, the coeloconic tormogen cell bears a strong resemblance to its basiconic counterpart. Both surround the subcuticular portion of outer dendrites with their apical processes, terminate their processes near the sensillum base, and extend a long, stalk-like basal protrusion (*Figure 11E* and *Video 8*). In future research, it will be interesting to determine whether the two coeloconic trichogen cells are functionally distinct, and how the morphological characteristics of individual auxiliary cells subserve function.

## Concluding remarks

We have presented a systematic morphological and morphometric analysis based on SBEM data sets of cryofixed *Drosophila* antennal tissues. In addition to providing nanoscale measurements of 33 identified ORN types and their corresponding sensillum apparatus, our analysis reveals several novel features of olfactory sensilla, including extracellular vacuoles, empty sensilla, and two previously uncharacterized basiconic sensillum types. Additionally, we find that not all at4 sensilla house three neurons; a subset contains only two ORNs. For most ORN types, neurons expressing the same receptor can display marked variability in their dendritic branching and morphology, suggesting that these features are not entirely genetically predetermined but are instead dynamically regulated. Moreover, contrary to the previously proposed paintbrush model, the basiconic outer dendrites exhibit a highly intricate branching pattern: diverging points are numerous, and branches can bifurcate or converge with others. Within basiconic sensilla, the large-spike ORNs selectively exhibit enlarged inner dendrites which are enriched with mitochondria. Furthermore, a remarkable morphological disparity is observed between compartmentalized ab2 neurons; the large-spike 'A' neuron sprouts more than 300 dendritic branches occupying nearly the entire sensillum lumen, whereas the neighboring 'B' neuron has only one unbranched outer dendrite.

These observations raise many interesting questions. For example, what is the functional significance of the extracellular vacuoles commonly found in the basiconic lumen? Does the mitochondria-enriched inner dendritic enlargement influence olfactory response properties? What molecules regulate the diverse outer dendritic morphology? Does the surface area disparity between neighboring neurons (e.g., ab2A vs. ab2B) or between neuronal types (e.g., ab2A vs. ac3A) contribute to differences in olfactory sensitivity? What are the molecular identities for neurons housed in the two novel basiconic sensilla? Do the empty sensilla represent snapshots of neuronal turnover and adult neurogenesis? Why are the coeloconic neurons supported by not one but two trichogen cells? In addition to raising new questions, the rich morphological and morphometric information provided here lays the foundation for future biophysical studies and computational modeling to determine the impact of neuronal size and shape on sensory responses, as well as on

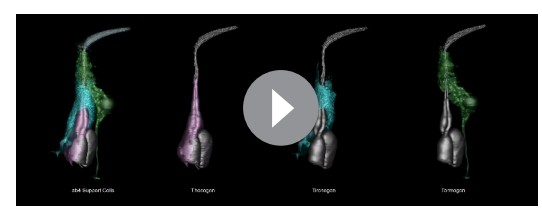

**Video 8.** 3D models of ab4 and ac3 auxiliary cells. Color codes are as indicated in *Figures 10* and *11*. Detailed information regarding individual 3D models shown in the videos can be found in the '*Table 2—source data 1*' file.

https://elifesciences.org/articles/69896#video8

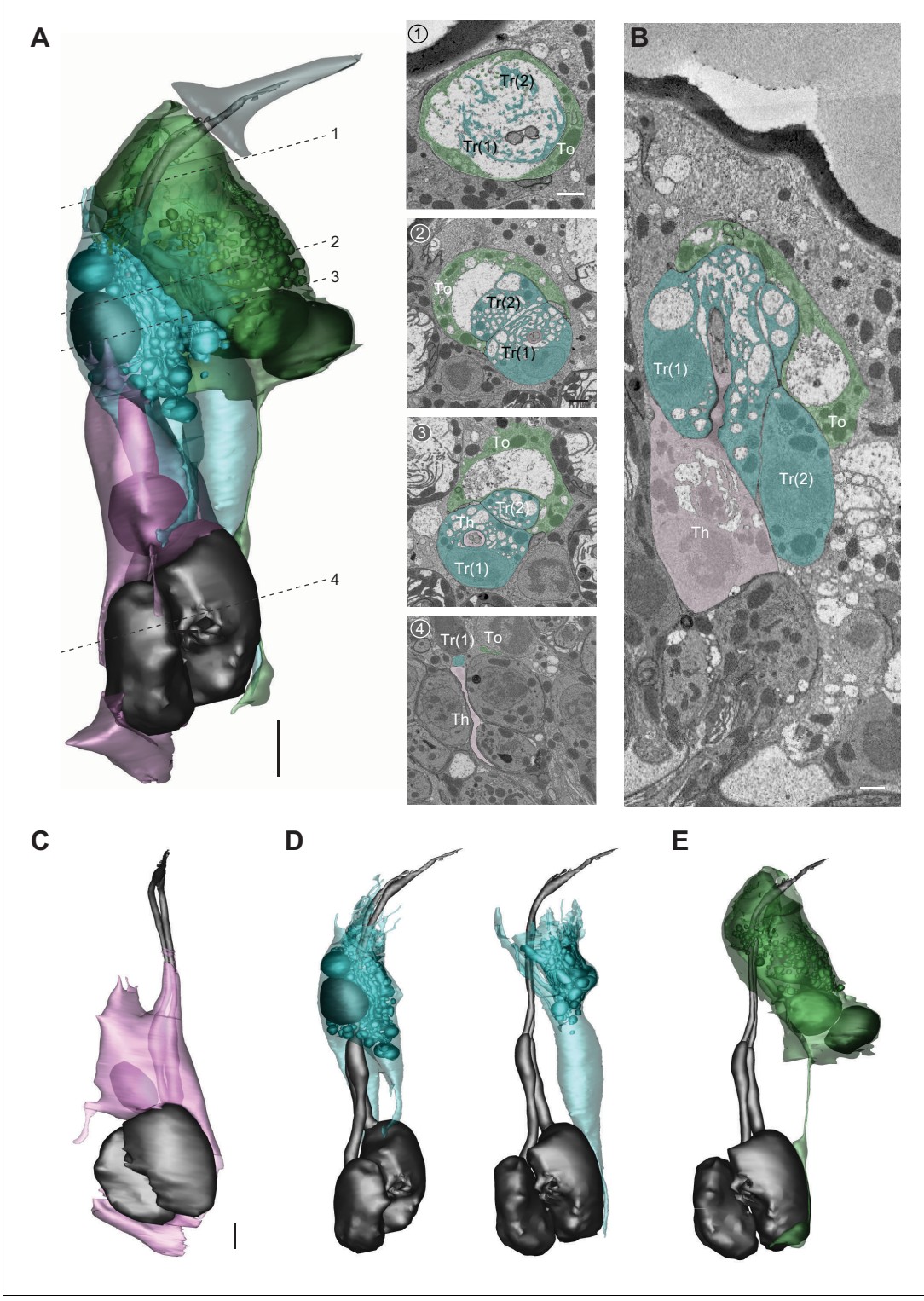

**Figure 11.** The four auxiliary cells for a coeloconic sensillum. (**A**) 3D model and SBEM images of a representative ac3 sensillum, highlighting its four auxiliary cells. Cells are pseudocolored to indicate identities: ORNs (gray), thecogen cell (pink), trichogen cell (turquoise), and tormogen cell (green). Dashed lines indicate positions of the corresponding SBEM images. (**B**) IMOD software-rendered SBEM image of a longitudinal sensillum section. Th: thecogen; Tr: trichogen; and To: tormogen. (**C–E**) 3D models of individual auxiliary cells as shown in (**A**): thecogen cell (**C**), trichogen cells, Tr(1) and Tr(2) (**D**), and tormogen cell (**E**). Scale bars: 2 μm for 3D models and 1 μm for

*Figure 11 continued on next page*

*Figure 11 continued*

SBEM images. The scale bar in the first image panel also pertains to other images unless indicated otherwise. ORN, olfactory receptor neuron; SBEM, serial block-face scanning electron microscopy.

ephaptic coupling between compartmentalized neurons.

## Materials and methods

### SBEM data sets
The SBEM data sets were generated previously (*Tsang et al., 2018*; *Zhang et al., 2019*), except for the Or7a-labeled data set. Briefly, 6–8-day-old females were cold anesthetized and their antennae removed. Following isolation of the third antennal segment, the sample was subjected to high-pressure freezing, freeze-substitution, rehydration, DAB labeling, en bloc heavy metal staining, dehydration, and resin infiltration. The flies expressed membrane-tethered APEX2 (*10xUAS-myc-APEX2-Orco* or *10xUAS-mCD8GFP-APEX2*) in select ORNs under the control of specific *OrX-* or *Ir-X GAL4* drivers as described (*Tsang et al., 2018*; *Zhang et al., 2019*). Following microcomputed X-ray tomography to determine the position and proper orientation of the resin-embedded specimens, samples were mounted on aluminum pins with conductive silver epoxy and sputter coated with gold-palladium for SBEM imaging with a Gemini SEM (Zeiss) equipped with a 3View block-face unit or a Merlin SEM (Zeiss) equipped with a 3View2XP and OnPoint backscatter detector. Detailed information regarding the image acquisition parameters can be found in the *Figure 1—source data 1* file. The SBEM image volumes are available in the Cell Image Library (http://www.cellimagelibrary. org/home). The accession numbers are CIL:54606 (Or22a/ab3A-labeled volume); CIL:54614 (Or7a/ ab4A-labeled volume); CIL:54610 (Or56a/ab4B-labeled volume); CIL:54611 (Ir75c/ac3AII-labeled volume); CIL:54612 (Or47a/ab5B-labeled volume); CIL:54607 and CIL:54608 (Or47b/at4A-labeled volumes); and CIL:54609 (Or88a/at4C-labeled volume).

### Isosurface models
Sensillum isosurface models were generated with the IMOD software (https://bio3d.colorado.edu/ imod/) (*Kremer et al., 1996*) using the 'Isosurface' function. Briefly, the 'bounding box parameters' were set to mark the regions of interest, thereby defining the subvolumes for isosurface rendering. Threshold values were manually adjusted based on image quality.

### Image segmentation
Segmentation was performed manually using the IMOD software. The 'drawing tools' function was used to trace and outline the structure of interest through serial sections of SBEM images. Surface models were generated using 'imodmesh' functions to connect closed contours whereby defined contours were connected with those directly above and below them. Further information on 'imodmesh' and meshing parameters can be found in the IMOD User's Guide (https://bio3d.colorado. edu/imod/doc/man/imodmesh.html). The cell body, inner dendrite, and individual outer dendritic branches were segmented as independent objects to allow for further morphometric analysis of these different cellular regions. In cases where the same structure occupied separate areas on a single SBEM section (e.g., looping or bending of a dendrite), the different parts of the same structure were segmented as different objects to allow their respective centroids to be extracted separately, thus ensuring accurate length measurement (see below).

### Morphometric analysis
The sensillum cuticle, ORN soma, inner, and outer dendritic segments were analyzed as separate objects. The inner and outer dendrites were demarcated by the ciliary constriction (*Shanbhag et al., 2000*). The 'imodinfo' function was used to extract morphometric data from surface models. We applied the '-c' option in the line command to determine volume and surface area for each region. Further information on 'imodinfo' can be found in the IMOD User's Guide (https://bio3d.colorado. edu/imod/doc/man/imodinfo.html). For branched outer dendrites, the volume and surface area for

the entire outer dendritic region were calculated by summing the volume or surface area of individual branches.

To determine the length of a sensillum, the 3D model was first smoothed using the IMOD function 'smoothsurf'. Centroid values for contours bounding the surface mesh were then extracted using the line command 'imodinfo' and the '-e' option. In the cases where the sectioning angle was oblique to the sensillum, the position of the centroid was visually verified or manually edited. Subsequently, the length of a sensillum was determined by summing the distances between sequential centroids.

## Skeletonization

The 3D models were converted to vrml2 files using the IMOD command 'imod2vrml2', imported into Amira software (2020.2 version; Thermo Fisher Scientific), and then scanned to a binary image volume such that every pixel of the model was indicated in white while the background in black. The AutoSkeleton module was used to skeletonize the binary image volume. These skeletons, in swc format, were imported into neuTube software (https://www.neutracing.com/) to allow branches to be manually spread out onto a 2D plane. Briefly, a primary branch and all its downstream branches were first selected to allow all the branches to be edited and moved as a group. This process was repeated for secondary, tertiary, and higher-order branches until overlap between branches was minimized.

The positions of ciliary constrictions and sensillum bases on the skeletons were determined by superimposing the spread-out skeletons on the original 3D model contours in swc format. First, the original contour skeleton was obtained by (1) extracting the coordinates of the points in the contours with the IMOD command 'model2point'; (2) converting the file to the csv format; (3) importing the csv file into R, and (4) connecting the points of each contour. Subsequently, the spread-out skeleton was imported into R and superimposed with the contour skeleton using the R package NeuroAnatomy Toolbox (Gregory S X E Jefferis and *Jefferis and Manton, 2017*; https://natverse.org/nat/). The combined skeleton and contours were exported in the swc format to be processed in neuTube where the ciliary constrictions and sensillum bases were manually annotated. The contour skeleton was then removed using NeuroAnatomy Toolbox to allow the annotated spread-out skeleton to be exported in neuTube for final image display.

## Statistics

All values were presented as mean± s.e.m. Paired two-tailed t-tests were performed for morphometric comparisons between ORNs housed in the same sensillum. Unpaired two-tailed t-tests were performed for comparisons between non-neighboring neurons. $P<0.05$ was taken as statistically significant.

## Acknowledgements

The authors thank Tin Ki Tsang for help with generating SBEM data sets, Isabel Nguyen, Andrea Nguyen, Alexandra Macaraeg, Valorie Gonzalez, and Kaitlyn Chu for assistance in image segmentation, and Larry Squire for comments on the manuscript. This study was supported by NIH grants R01DC016466, R01DC015519, and R21DC108912 (C-YS); U24NS120055, R24GM137200, and R01GM082949 (MHE). The authors dedicate this study to Prof. Dr. RA Steinbrecht, whose landmark papers have profoundly influenced our research.

## Additional information

### Funding

| Funder | Grant reference number | Author |
| --- | --- | --- |
| National Institute on Deafness and Other Communication Disorders | R01DC016466 | Chih-Ying Su |
| National Institute on Deafness and Other Communication | R01DC015519 | Chih-Ying Su |

| | | | |
|---|---|---|---|
| Disorders | | | |
| National Institute on Deafness and Other Communication Disorders | R21DC108912 | | Chih-Ying Su |
| National Institute of Neurological Disorders and Stroke | U24NS120055 | | Mark H Ellisman |
| National Institute of General Medical Sciences | R24GM137200 | | Mark H Ellisman |
| National Institute of General Medical Sciences | R01GM082949 | | Mark H Ellisman |
| National Science Foundation | NSF2014862-UTA20-000890 | | Mark H Ellisman |

The funders had no role in study design, data collection and interpretation, or the decision to submit the work for publication.

### Author contributions

Cesar Nava Gonzales, Conceptualization, Data curation, Formal analysis, Validation, Investigation, Visualization, Methodology; Quintyn McKaughan, Conceptualization, Data curation, Formal analysis, Validation, Investigation, Visualization, Methodology, Writing - original draft, Writing - review and editing; Eric A Bushong, Software, Investigation, Methodology, Writing - review and editing; Kalyani Cauwenberghs, Software, Formal analysis, Visualization, Methodology, Writing - original draft, Writing - review and editing; Renny Ng, Investigation, Visualization, Writing - original draft, Writing - review and editing; Matthew Madany, Software, Methodology; Mark H Ellisman, Resources, Funding acquisition; Chih-Ying Su, Conceptualization, Resources, Data curation, Formal analysis, Supervision, Funding acquisition, Validation, Investigation, Visualization, Writing - original draft, Project administration, Writing - review and editing

### Author ORCIDs

Quintyn McKaughan ⬚ https://orcid.org/0000-0001-6611-6530
Chih-Ying Su ⬚ https://orcid.org/0000-0002-0005-1890

### Decision letter and Author response

Decision letter https://doi.org/10.7554/eLife.69896.sa1
Author response https://doi.org/10.7554/eLife.69896.sa2

## Additional files

### Supplementary files

- Transparent reporting form

### Data availability

All data generated or analyzed during this study are included in the manuscript and supporting files. Source data files have been provided for Table 1, Table 2 and Individual datasets. The SBEM image volumes are available in the Cell Image Library (http://www.cellimagelibrary.org/home). The accession numbers are CIL:54606 (Or22a/ab3A-labeled volume); CIL:54614 (Or7a/ab4A-labeled volume); CIL:54610 (Or56a/ab4B-labeled volume); CIL:54611 (Ir75c/ac3AII-labeled volume); CIL:54612 (Or47a/ab5B-labeled volume); CIL:54607 and CIL:54608 (Or47b/at4A-labeled volumes); CIL:54609 (Or88a/at4C-labeled volume).

The following datasets were generated:

| Author(s) | Year | Dataset title | Dataset URL | Database and Identifier |
|---|---|---|---|---|
| Tsang TK, Bushong EA, Ellisman MH, Su C-Y | 2021 | Or88a-GAL4>10XUAS-myc-APEX2-Orco | http://www.cellimagelibrary.org/images/54609 | Cell Image Library, CIL:54609 |

| Tsang TK, Bushong EA, Ellisman MH, Su C-Y | 2021 | Or47b-GAL4>10XUAS-myc-APEX2-Orco | http://www.cellimageli-brary.org/images/54608 | Cell Image Library, CIL:54608 |
|---|---|---|---|---|
| Tsang TK, Bushong EA, Ellisman MH, Su C-Y | 2021 | Or47b-GAL4>10XUAS-myc-APEX2-Orco | http://www.cellimageli-brary.org/images/54607 | Cell Image Library, CIL:54607 |
| Tsang TK, Bushong EA, Ellisman MH, Su C-Y | 2021 | Or47a-GAL4>10XUAS-myc-APEX2-Orco | http://www.cellimageli-brary.org/images/54612 | Cell Image Library, CIL:54612 |
| Tsang TK, Bushong EA, Ellisman MH, Su C-Y | 2021 | Ir75c-GAL4>10XUAS-myc-APEX2-Orco | http://www.cellimageli-brary.org/images/54611 | Cell Image Library, CIL:54611 |
| Tsang TK, Bushong EA, Ellisman MH, Su C-Y | 2021 | Or56a-GAL4> 10XUAS-myc-APEX2-Orco | http://www.cellimageli-brary.org/images/54610 | Cell Image Library, CIL:54610 |
| Tsang TK, Bushong EA, Ellisman MH, Su C-Y | 2021 | Or7a-GAL4>10XUAS-myc-APEX2-Orco | http://www.cellimageli-brary.org/images/54614 | Cell Image Library, CIL:54614 |
| Tsang TK, Bushong EA, Ellisman MH, Su C-Y | 2021 | Or22a-GAL4> 10XUAS-mCD8GFP-APEX2 | http://www.cellimageli-brary.org/images/54606 | Cell Image Library, CIL:54606 |

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
