## [Decision Letter]

**Acceptance summary:**

The editors and reviewers were impressed by the breath and depth of the work, which will serve as the basis for future studies on the function, development and evolution of sensory sensilla of the fruit fly's antennae. We were very pleased that your data set was deposited into a public repository (Cell Image Library), which is key to elevate your findings to the status of a reference in the field, enabling others to build on your work in the best scientific tradition.

**Decision letter after peer review:**

Thank you for submitting your article "Systematic morphological and morphometric analysis of identified olfactory receptor neurons in *Drosophila melanogaster*" for consideration by *eLife*. Your article has been reviewed by 2 peer reviewers, and the evaluation has been overseen by a Reviewing Editor and Piali Sengupta as the Senior Editor. The following individual involved in review of your submission has agreed to reveal their identity: Jürgen Rybak (Reviewer #3).

Essential revisions:

1) Please clarify and detail how the identification of each sensilla was done. Please also be careful to avoid circular arguments regarding the identification of a sensillum and the use of that very set of characteristics for identification also for the comparative analysis across sensilla.

2) Please clarify how the data will be publicized and distributed. This data set is exceptional and enables further either follow up studies or orthogonal studies on a variety of aspects of sensilla morphology and its relationship to function.

3) Please attempt to address the detailed comments listed by the reviewers, as these capture a number of small issues whose resolution will only strengthen and clarify your beautiful findings.

*Reviewer #2 (Recommendations for the authors):*

Generally, given the descriptive nature of the manuscript, it would be good to include when possible graphs showing correlations between the identified morphological features and published functional data. I think this could add to the interpretation of their findings and give further insights.

I find the discussion could be improved by adding ideas on the possible functional implications of some of their key findings. For example, for the role of mitochondria and membranous protrusions.

Do the authors think that the basal vacuoles in trichoid sensilla are the same type as those distal in large basiconic sensilla? A discussion on these in the Discussion section would be welcomed.

The description of the auxiliary cells is a bit dry and for those with little knowledge on the presumed or demonstrated function of auxiliary cells difficult to interpret. I would recommend adding a bit of background on auxiliary cells and discuss the possible implications of their findings in the discussion.

Finally, as mentioned above, I would like to insist that given the focus of the manuscript, it is key that the authors provide a better description for the ways in which sensilla were identified, and discuss any drawbacks of their approach. This would include the suggested schematics in Figure 1 showing from which region in the antenna the datasets were collected. Furthermore, I think it is important to include in the figure legends and/or methods information on the dataset of origin for each of the sensilla used for their analysis.

*Reviewer #3 (Recommendations for the authors):*

General comments to text and figures

– The extensive dataset potentially allows to study in more detail the ultrastructure and other cellular features of the ORNs, and its support cells. Therefore, it would be very valuable to make available the full SBEM datasets for the neuroscience community in an open repository, e.g.(https://github.com/). Suggestions on implementation software (e.g. https://github.com/catmaid/CATMAID) would be very helpful.

– Provide a more comprehensive table with respect to your identified sensilla/ORN types.

– How large is the variability of location of (identified sensilla) on the antenna?

– Throughout the text, please give always the number, (n=), consistently for the number of ORN/sensilla to avoid cross-reading. For the non-specialized reader parts of your results are had to follow.

– Is ciliary constriction a generic feature in all sensilla?

– The thecogen cell, which is itself enveloped by the trichogen processes and then further surrounded by the tormogen cell. Please, provide a proper introduction, and discussion on these cell types. Does your study provide more insight into their function see, also, Larter, N. K., et al.,(2016). "Organization and function of *Drosophila* odorant binding proteins." *eLife* 5: e20242.

– The inner dendrite of a atypical neuron ac1 contained 24 mitochondria. Is this a single observation found in one neuron? You described mito-enriched inner dendrites in other ORN types: are there structural correlations to other parts of the dendrite or might is simply a plasticity based change?

Please, give a summary statement of the outcome of your statistical analysis: how much variability, how much generic, features do you find for the different ORN classes. Is soma size correlated with the dendritic branching in the outer dendrite?

– Abstract

– state the number of sensilla studied here

Introduction

– introduce the cell types thecogen, tormogen trichogen, use citations, what are the possible roles of these supporting cells?

– 1st para dendritic branching pattern? are these known from one of the authors cited above?

– there is evidence that centrifugal cells might modulates ORN response properties. Did you find morphological correlates of electrical or chemical synapses in your dataset? For example, the antenna is innervated by a large centrifugal aminergic neuron in other insect species (Schroter et al., 2007). Wang, J. W. (2012). Presynaptic modulation of early olfactory processing 10.1002/dneu.20936 Jung, J. W., Kim, (2013). Neuromodulation of Olfactory …..the American Cockroach, Periplaneta americana.

– page 4: 33 ORN types out of how many types of sensilla?

– page 4, 541 sensilla how many for each sensilla type out of 541 sensilla? Are these sampled from one female *Drosophila* antenna, or did you use several specimen. In Keesey (2019) the total number of sensilla are reported as to app. 400?

– page 4/5- characteristic topographical distribution: please refer in the text to Figure 1a and specify, in detail of how you determined the relative position of the sensillum of interest. see also my comments for Figure 1.

– sensillum identity? Criteria for the identification of non-genetically labeled and identified sensilla should be explained in more detail.

page 6 but some terminate before reaching the endpoint (Figure 2A), what is the diameter of the dendrite? do you mean branch, branch versus dendrite!!

page 6: provide larger image of the vacuoles., what lumen is inner or outer dendrite.

page 7: the info on T2 is insufficient ….

page 9: dendrite is unbranched?

Method

– Explain technical terms, and the technical workflow, so that the non-specialized reader can follow.

– More details on SBEM methodology, how is the quality of DAB labelling – is the DAB staining cytoplasmic, or membrane-bound – what resolution you used for sectioning?

– How did you perform the visualization of pores across the sensillum surface, from the EM images, or did you perform a SEM scan beforehand.

– Is imod an open software – specify the version of Amira.

– Isosurface is not suited for volume measure, how was the surface model generated and smoothed?

– Morphometry: how did you created the surface models, explain imodinfo..

Centroid values are a skeleton tool?

– Ball-and stick model?

Results

In general: for the text on different sensilla types in each sub-chapter, I suggest to always mention the corresponding ORN receptor name (and vice versa), for the convenience and less cross reading. Even though the data are summarized in Table 1 the non-specialized reader is not used to the terminology.

For example page 12, ab4 sensillum, the two ab4 neuron, ab4A and B should be named, and indicated, as Or7a and Or56a, respectively. Same for the legends.

– Is there always an positive correlation of soma size with dendritic arborization of the respective ORN?

Discussion

Please, discuss and cite the relevant literature related your novel findings in more detail. how the size and shape of sensory neurons influence their responses, sensitivity and circuit function of ORNs?

– New receptors found, e.g. ABx. Give suggestions of how can these be identified (in situ hybridization?)

– Some OSNs express more than one receptor. see the work by the Potter lab: Task (2020) Widespread Polymodal Chemosensory Receptor Expression in *Drosophila* Olfactory Neurons 2020.11.07.355651 for discussion discussed in this context.

The ORN in ab4b expresses two receptors: Or56a that respond specifically to goesomin and Or33a that no one could identify its ligand yet.

– How often did you find extracellular vacuoles within the sensilla lumen, in how many sensilla types they occur?

– Intricate dendritic branching in the outer dendrite ab4A neuron: Figure 8C, and text

In the 2D projection dendritic arbors seems to be fused, that contradicts neuron dichotomy morphology. Is this a simple crossover, or a dendritic tiling shown, here? as described in *Drosophila* larva Grueber (2002).:Tiling of the *Drosophila* epidermis Development, 129(12), see Figure 2B, here. Does the dendritic morphology correlates with arrangement or number of pores in the respective sensillum.

– Empty sensilla containing no neurons, are they described before?

Could this phenomen due to a cell turnover, or apoptosis? using P-MARCM Fernández-Hernández (2020). Reported a 3% turnover of ORNs.

Concluding remarks

Discuss in more detail your question with the literature what is known. Are empty cells reported before? How your data relate to other studies on the antenna.e.g. Keesey (2019) Inverse location https://www.nature.com/articles/s41467-019-09087-z, for the distribution of sensilla on the antenna.

Figures

– Figure 1A: the distribution on Apex labelled flies (false color?). Did you sample from the anterior / posterior side of the antenna? Please give an orientation, and a relative position of the sampled sensilla, e.g. location of arista, sacculus, and funiculus border (for an nice example, see e.g. Grabe (2016) Figure 1B).

– 1B: the columns are show the number of identified sensilla, I suggest, to Indicate in Figure 1 A with stars the labelled versus non-labeled sensilla,

– In Figure 1 you gave an overview of datasets used in the study. Please, indicate: how large is your dataset, i.e 123 ORNs, relative to the total number of sensilla (Keesey 2019) reported app. 400 sensilla in total for *Drosophila* m. Also: are your samples randomly chosen across the antenna.Is there a distinct pattern of how are sensilla types are distributed across the antenna for certain sensilla types ?

– Figure 2 the extracellular vacuole (in teal) is shown in dark blue in 2B (3)? I suggest to use another color for the vacuoles for better understanding of the 3D structure.

– Table 2 (main text): I suggest to provide more details for the identified ORN, to compare with previous datasets, or data repositories (such as DoOR) and to allow for more cross-references found in the literature: – OR co-receptor, – corresponding glomerulus in the antennal lobe, – functional response according to DoOR

In the DoOR database (http://neuro.uni-konstanz.de/DoOR/default.html ), for example, the ab4 sensillum hosing Or7A receptor (ab4A neuron), and the Or56a co-expressed with Or33Aa. The latter is not listed in your table.

Supplementary

Su_28-04-2021-RA-*eLife*-69896_Source_Data_File excel Table

In general: provide an explanation and an list of all abbreviation used in these excel files. There are partly hard to understand.

– ORN Morphometric: I do not understand the terminology of column 2.

– T3 model: why you present data three soma sizes: A,B,C, but listing only two ORN types in the first column ? same for AC1 model.

– It took me a while to figure out that the first column 'just' indicates the dataset (genetically labelled with Apex) without any other reference to the following columns identity of ORN, and labelled with A,B, or C.

I suggest to present, here, and in the tables where you report 'branching' and 'mitochondria' the ORN type in brackets, as you do in the Summary Table.

Videos: Please, prepare an extra sheet, word-file to provide a detailed description for each video.

---

## [Author Response]

Essential revisions:1) Please clarify and detail how the identification of each sensilla was done. Please also be careful to avoid circular arguments regarding the identification of a sensillum and the use of that very set of characteristics for identification also for the comparative analysis across sensilla.

We thank the editor and reviewers for raising this point, in particular regarding the distinction between ab2 and ab3 sensilla. In the Results, we have now clarified that “Among the two large basiconic sensilla that house two neurons, we distinguished ab2 from ab3 by its lack of DAB staining in the Or22a dataset, in which ab3A was genetically labeled by APEX2. Apart from the Or22a dataset, an ab2 sensillum was identified in the Or7a dataset on the basis of its proximity to the labeled ab4 sensilla, because ab3 is not found in the same topographical region as ab4 (de Bruyne et al., 2001).”

We have also indicated how each sensillum type was identified in the revised Source Data for Table 1.

2) Please clarify how the data will be publicized and distributed. This data set is exceptional and enables further either follow up studies or orthogonal studies on a variety of aspects of sensilla morphology and its relationship to function.

We thank the editor and reviewers for raising this critical point. All eight SBEM image volumes described in this study have been deposited in the Cell Image Library (http://www.cellimagelibrary.org/home). The accession numbers are provided in the revised “SBEM datasets” under the Materials and methods section.

3) Please attempt to address the detailed comments listed by the reviewers, as these capture a number of small issues whose resolution will only strengthen and clarify your beautiful findings.

We appreciate the opportunity to improve the clarity of our manuscript. Please see below a summary of the key revisions.

– We have included a new panel (Figure 1C) to illustrate the antennal regions covered by individual SBEM volumes.

– We have expanded Table 1 to include the mitochondria number and glomerular projection information.

– The vacuole color has been changed from teal to aubergine to better distinguish this structure from others.

– We have added the statistical analysis results for morphometric comparison in the figure legends.

– We have provided background information regarding the function of olfactory auxiliary cells.

– We have provided a new Source Data file which details the image acquisition parameters for individual datasets. In the same file, we also indicated the identity of individual sensilla identified in each dataset.

– We have expanded the previous two Source Data files to provide additional information requested by the reviewers, including the sensillum identification criteria, 3D models shown in the main figures and videos, and spike amplitude ratios.

– We have provided a more detailed description of the methods used in our study, including the links to different features we employed with the IMOD software.

Reviewer #2 (Recommendations for the authors):Generally, given the descriptive nature of the manuscript, it would be good to include when possible graphs showing correlations between the identified morphological features and published functional data. I think this could add to the interpretation of their findings and give further insights.

We thank the reviewer for the suggestion. We wish to clarify that the odor response profile of an ORN is predominantly determined by the receptor expressed in the neuron (Hallem et al., Cell, 2004), which is independent of whether the neuron is a large- or small-spike ORN. By extension, the odor response profile is independent of the neuron’s morphometric features. It is therefore of limited usefulness to search for any correlation between ORNs’ morphological features and odor response properties.

However, we have incorporated the reviewer’s suggestion by revising the text to include key ligands for certain ORNs that respond to ethologically salient odors. In addition, we have added the following sentence in the revised Table 2 legend: “The odor response profiles for many of the characterized ORNs can be found in the DoOR database (http://neuro.uni-konstanz.de/DoOR/default.html)” such that readers who are curious about the functional data can easily find the information.

I find the discussion could be improved by adding ideas on the possible functional implications of some of their key findings. For example, for the role of mitochondria and membranous protrusions.

We thank the reviewer for the suggestion. As the reviewer pointed out, our study is descriptive in nature, aiming to provide an extensive morphological and morphometric dataset for identified *Drosophila* ORNs. We hope our findings will stimulate future investigations into the developmental mechanism and functional significance of the rich diversity in ORN size and shape. As such, we prefer to highlight the multitude of intriguing questions, instead of speculating on the answers in our concluding remarks.

Nevertheless, we agree with the reviewer that we could provide more information on mitochondria function in olfactory signaling. In the revised Results, we have now included “We note that in vertebrate ORNs, mitochondria play a direct role in regulating cytosolic ca^2+^ response profile and thereby ensure a broad dynamic range for the neurons’ spike responses (Fluegge et al., 2012). Although it is unclear whether mitochondria play a similar role in insect olfactory signaling, a recent study shows that odor-induced ca^2+^ signals in *Drosophila* ORNs are shaped by mitochondria (Lucke, Kaltofen, Hansson, and Wicher, 2020). Therefore, it will be interesting to investigate the functional significance of this striking mitochondrial disparity between grouped ORNs in future research” under the “Large Basiconic Sensilla” section.

Do the authors think that the basal vacuoles in trichoid sensilla are the same type as those distal in large basiconic sensilla? A discussion on these in the Discussion section would be welcomed.

We thank the reviewer for the careful reading of our manuscript. In this study, we define vacuoles as extracellular membranous structures found in the sensillum lumen. In this context, we do not distinguish vacuoles found in different sensillum types. We have, however, incorporated the reviewer’s comment by adding this sentence:

“Future research may be directed to determine whether the location of vacuoles influences their function.”

The description of the auxiliary cells is a bit dry and for those with little knowledge on the presumed or demonstrated function of auxiliary cells difficult to interpret. I would recommend adding a bit of background on auxiliary cells and discuss the possible implications of their findings in the discussion.

We thank the reviewer for the excellent suggestion. We have now provided background information on olfactory auxiliary cells in the revised Results, which reads “Auxiliary cells support olfactory function by insulating ORN soma and inner dendrites, maintaining transepithelial potential, expressing ion transporters, and secreting odorant binding proteins and odorant degrading enzymes into the sensillum lymph (Kaissling, 1986; Larter, Sun, and Carlson, 2016; Leal, 2013; Menuz, Larter, Park, and Carlson, 2014; Shanbhag et al., 2001, 2000; Sun, Xiao, Carlson, and Carlson, 2018)”.

We have also included the sentence “In future research, it will be interesting to determine whether the two coeloconic trichogen cells are functionally distinct, and how the morphological characteristics of individual auxiliary cells subserve function” at the end of the Olfactory Auxiliary Cells section.

Finally, as mentioned above, I would like to insist that given the focus of the manuscript, it is key that the authors provide a better description for the ways in which sensilla were identified, and discuss any drawbacks of their approach. This would include the suggested schematics in Figure 1 showing from which region in the antenna the datasets were collected. Furthermore, I think it is important to include in the figure legends and/or methods information on the dataset of origin for each of the sensilla used for their analysis.

We thank the reviewer for the helpful suggestions. We have added the suggested schematics in Figure 1 illustrating the regions covered by individual SBEM volumes. Further, we have now provided detailed information regarding how each sensillum was identified in the text and indicated its source dataset in the Source Data for Table 1. In addition, in the Source Data for Table 2, we have also indicated the 3D ORN models shown in the main and supplementary Figures.

Reviewer #3 (Recommendations for the authors):General comments to text and figures– The extensive dataset potentially allows to study in more detail the ultrastructure and other cellular features of the ORNs, and its support cells. Therefore, it would be very valuable to make available the full SBEM datasets for the neuroscience community in an open repository, e.g.(https://github.com/). Suggestions on implementation software (e.g. https://github.com/catmaid/CATMAID) would be very helpful.

We thank the reviewer for raising this critical point. All eight SBEM image volumes described in this study have been deposited in the Cell Image Library (http://www.cellimagelibrary.org/home). We have also provided the accession numbers in the revised “SBEM datasets” under the Materials and methods section.

–Provide a more comprehensive table with respect to your identified sensilla/ORN types.

As suggested, we have included ORN glomerulus projection as an additional identifier in Table 2. The revised legend now includes:

“ORN identity is indicated by the sensillum type, relative spike amplitude (A, B, C or D), odor-tuning receptor, and glomerular projection.”

– How large is the variability of location of (identified sensilla) on the antenna?

We wish to clarify that our study focuses on the nanoscale morphological and morphometric features of sensilla and ORNs, instead of the distribution of sensilla on the antenna, which have been thoroughly described in multiple studies. We note that each SBEM volume sampled a specific portion of the antenna covering the APEX2-labeled sensilla (Figure 1C), making it difficult to precisely determine its relative antenna location or orientation. Therefore, we cannot comment on the variability of sensilla location among different SBEM datasets.

– Throughout the text, please give always the number, (n=), consistently for the number of ORN/sensilla to avoid cross-reading. For the non-specialized reader parts of your results are had to follow.

We thank the reviewer for the suggestion. The numbers of the 3D models are available in the corresponding figure legends, as well as in Table 2.

– Is ciliary constriction a generic feature in all sensilla?

Yes, ciliary constriction is a characteristic feature of insect ORNs. This constriction demarcates the inner and outer dendrites. We have also added the reference (Shanbhag et al., 1999) in the revised text.

– The thecogen cell, which is itself enveloped by the trichogen processes and then further surrounded by the tormogen cell. Please, provide a proper introduction, and discussion on these cell types. Does your study provide more insight into their function see, also, Larter, N. K., et al.,(2016). "Organization and function of *Drosophila* odorant binding proteins." eLife 5: e20242.

We thank the reviewer for the excellent suggestion. We have now provided background information on olfactory auxiliary cells in the revised Results, which reads:

“Auxiliary cells support olfactory function by insulating ORN soma and inner dendrites, maintaining transepithelial potential, expressing ion transporters, and secreting odorant binding proteins and odorant degrading enzymes into the sensillum lymph (Kaissling, 1986; Larter, Sun, and Carlson, 2016; Leal, 2013; Menuz, Larter, Park, and Carlson, 2014; Shanbhag et al., 2001, 2000; Sun, Xiao, Carlson, and Carlson, 2018)”.

We have also included the sentence:

“In future research, it will be interesting to determine whether the two coeloconic trichogen cells are functionally distinctive, and how the morphological characteristics of individual auxiliary cells subserve function.”

at the end of the Olfactory auxiliary Cells section.

– The inner dendrite of a atypical neuron ac1 contained 24 mitochondria. Is this a single observation found in one neuron?

Yes, this is a single observation as indicated in the text and Supplementary figure.

You described mito-enriched inner dendrites in other ORN types: are there structural correlations to other parts of the dendrite or might is simply a plasticity based change?

The reviewer raised an interesting question. However, given that our method does not allow us to image the same ORNs longitudinally over time, it is difficult for us to determine whether the feature is plastic or not.

Please, give a summary statement of the outcome of your statistical analysis: how much variability, how much generic, features do you find for the different ORN classes. Is soma size correlated with the dendritic branching in the outer dendrite?

We thank the reviewer for the suggestion. We have now included statistical analysis results in the figure legends when the information is available.

Regarding soma size and dendritic branching, it is unclear whether the two features are correlated. Dendritic arborization depends primarily on the ORN morphological class. As described in our manuscript, for a given ORN type (T1 for example), we observed noticeable variability in dendritic arborization pattern among T1 ORNs despite their similar soma size.

– Abstract– state the number of sensilla studied here

In consideration of the abstract word limit, we instead provided this information in the last paragraph of the Introduction. A total of 158 3D sensillum models were reconstructed for this study. Detailed information can also be found in the beginning of the Results section:

“Among the whole sensilla, we identified 37 basiconic, 30 coeloconic, 40 intermediate and 51 trichoid sensilla based on their distinctive morphological features”.

Introduction– introduce the cell types thecogen, tormogen trichogen, use citations, what are the possible roles of these supporting cells?

We thank the reviewer for the excellent suggestion. We have now provided background information on olfactory auxiliary cells in the revised Results, which reads:

“Auxiliary cells support olfactory function by insulating ORN soma and inner dendrites, maintaining transepithelial potential, expressing ion transporters, and secreting odorant binding proteins and odorant degrading enzymes into the sensillum lymph (Kaissling, 1986; Larter, Sun, and Carlson, 2016; Leal, 2013; Menuz, Larter, Park, and Carlson, 2014; Shanbhag et al., 2001, 2000; Sun, Xiao, Carlson, and Carlson, 2018)”.

– 1st para dendritic branching pattern? are these known from one of the authors cited above?

We apologize that our writing was not clear to the reviewer. We have indicated that “…we examined the branching pattern of ab4 outer dendrites and found it to be strikingly complex, contrary to the simple paintbrush model proposed on the basis of TEM longitudinal-section images of a sensillum, whereby all basiconic dendritic branches share a common diverging point (Shanbhag et al., 1999)

– There is evidence that centrifugal cells might modulates ORN response properties. Did you find morphological correlates of electrical or chemical synapses in your dataset? For example, the antenna is innervated by a large centrifugal aminergic neuron in other insect species (Schroter et al., 2007). Wang, J. W. (2012). Presynaptic modulation of early olfactory processing 10.1002/dneu.20936 Jung, J. W., Kim, (2013). Neuromodulation of Olfactory …..the American Cockroach, Periplaneta americana

We thank the reviewer for raising this interesting point. However, we wish to clarify that our study concerns only the antenna and does not cover the antennal lobe, the only location where presynaptic modulation of ORN axon terminals can occur.

To answer the reviewer’s question, we did not observe any synaptic structures in our SBEM datasets, consistent with the previous findings based on high-resolution TEM images of cryofixed *Drosophila* antenna (Shanbhag et al., 2000).

– page 4: 33 ORN types out of how many types of sensilla?

The information can be found in the last paragraph of Introduction:

“we reconstruct 3D structures for neurons housed in 13 of the 19 genetically defined antennal sensillum types”.

– Page 4, 541 sensilla how many for each sensilla type out of 541 sensilla? Are these sampled from one female *Drosophila* antenna, or did you use several specimen. In Keesey (2019) the total number of sensilla are reported as to app. 400?

We are sorry that our writing was not clear to the reviewer. We have now clarified this confusion in the revised Results

“Individual datasets, acquired from different flies and named on the basis of the APEX2-expressing ORNs, included numerous sensory hairs in addition to the sensilla that housed labeled neurons (Figure 1A). In agreement with the characteristic topographical distribution of sensilla on the antenna (de Bruyne, Foster, and Carlson, 2001; Grabe et al., 2016; Shanbhag et al., 1999), the four morphological sensillum classes were unevenly represented in our eight SBEM datasets (Figure 1B,C). A total of 541 sensilla were sampled from these datasets, comparable to the total number of sensilla on an antenna (Shanbhag et al., 1999). Among the whole sensilla, we identified 37 basiconic, 30 coeloconic, 40 intermediate and 51 trichoid sensilla based on their distinctive morphological features (Shanbhag et al., 1999).”

– Page 4/5- characteristic topographical distribution: please refer in the text to Figure 1a and specify, in detail of how you determined the relative position of the sensillum of interest. see also my comments for Figure 1.

We wish to clarify that our study focuses on the nanoscale morphological and morphometric features of sensilla and ORNs, instead of the distribution of sensilla on the antenna. Each SBEM volume sampled a specific portion of the antenna covering the APEX2-labeled sensilla, making it difficult to precisely determine its relative antenna location and orientation.

However, we note that we did not notice any major difference in the sensillum distribution pattern between our datasets and published results, as shown in Figure 1A.

– Sensillum identity ? Criteria for the identification of non-genetically labeled and identified sensilla should be explained in more detail.

As suggested, we have indicated how each sensillum type was identified in the revised text and Source Data for Table 1.

Page 6 but some terminate before reaching the endpoint (Figure 2A), what is the diameter of the dendrite? do you mean branch, branch versus dendrite!!

We are sorry for the confusion. “Branch” referred to dendritic branch.

Page 6: provide larger image of the vacuoles.,

Please see Figure 2B for an enlarged image of the vacuoles.

what lumen is inner or outer dendrite.Page 7: the info on T2 is insufficient ….Page 9: dendrite is unbranched?

We are sorry but we do not understand these questions.

Method– Explain technical terms, and the technical workflow, so that the non-specialized reader can follow.

This information can be found in the “SBEM datasets” section:

“Following isolation of the third antennal segment the sample was subjected to high-pressure freezing, freeze-substitution, rehydration, DAB labeling, en bloc heavy metal staining, dehydration, and resin infiltration.”

– More details on SBEM methodology, how is the quality of DAB labelling – is the DAB staining cytoplasmic, or membrane-bound – what resolution you used for sectioning?

DAB staining is membrane-bound. We have now revised the section to read:

“The flies expressed membrane-tethered APEX2 (*10xUAS-myc-APEX2-Orco* or *10xUAS-mCD8GFP-APEX2*) in select ORNs under the control of specific *OrX-* or *Ir-X GAL4* drivers as described (Tsang et al., 2018; Zhang et al., 2019).”

Detailed information of the methodology is available in a standalone method paper, which we published in 2018 (Tsang et al., *eLife*, 2018).

To address the reviewer’s question about resolution, we have provided a new “Source Data for Figure 1” file to detail imaging acquisition parameters.

– how did you perform the visualization of pores across the sensillum surface, from the EM images, or did you perform a SEM scan beforehand.

No, we did not perform SEM imaging for this study. In certain datasets, we could visualize pores on the sensillum surface via performing isosurface reconstruction using IMOD. This information is also available in the Figure 1 legend.

– Is imod an open software – specify the version of Amira.

Yes, we have now included the link for software download (https://bio3d.colorado.edu/imod/) in the revised Methods.

– Isosurface is not suited for volume measure, how was the surface model generated and smoothed?

The reviewer is correct that isosurface is not suited for volume measurement. However, we did not measure sensillum morphometrics based on isosurface images. Instead, sensillum volumes were determined based on the 3D models using the “imodinfo” function in IMOD, as indicated in the “Morphometric analysis” section.

The sensillum 3D surface images were generated using the “Isosurface” function in IMOD instead. This information is available in the “Isosurface models” section.

We have also explained in greater detail how surface models were generated “Surface models were generated using “imodmesh” functions to connect closed contours whereby defined contours were connected with those directly above and below them. Further information on “imodmesh” and meshing parameters can be found in the IMOD User’s Guide (https://bio3d.colorado.edu/imod/doc/man/imodmesh.html).”

– Morphometry: how did you created the surface models, explain imodinfo..

To address the reviewer’s comment, we have expounded on this point in the methods section, including:

“The ‘drawing tools’ plugin was used to trace and outline the structure of interest through serial section of SBEM images. Surface models were generated using ‘imodmesh’ functions to connect adjacent closed contours. Further information on ‘imodmesh’ and meshing parameters can be found in the IMOD User’s Guide (https://bio3d.colorado.edu/imod/doc/man/imodmesh.html)”.

Imodinfo is an IMOD function that allows for extraction of various morphometric data from surface models. Different line-command options extract different data, such as: mesh volume, mesh surface area, and centroid values. To make this clear, we have included in the methods, “The ‘imodinfo’ function was used to extract morphometric data from surface models. We applied option ‘-c’ in the line-command to determine volume and surface area for each region. Further information on ‘imodinfo’ and meshing parameters can be found in the IMOD User’s Guide (https://bio3d.colorado.edu/imod/doc/man/imodinfo.html)”.

Centroid values are a skeleton tool?

Centroid values were extracted using the IMOD function “imodinfo”. Centroid values represent the center of mass for contours bounding the surface models. Further information on ‘imodinfo’ and meshing parameters can be found in the IMOD User’s Guide (https://bio3d.colorado.edu/imod/doc/man/imodinfo.html).

– Ball-and stick model?ResultsIn general: for the text on different sensilla types in each sub-chapter, I suggest to always mention the corresponding ORN receptor name (and vice versa), for the convenience and less cross reading. Even though the data are summarized in Table 1 the non-specialized reader is not used to the terminology.For example page 12, ab4 sensillum, the two ab4 neuron, ab4A and B should be named, and indicated, as Or7a and Or56a, respectively. Same for the legends.

Per reviewer’s suggestion, we have now provided the receptor information in the main text for all previously characterized ORNs.

– Is there always an positive correlation of soma size with dendritic arborization of the respective ORN?

Regarding soma size and dendritic branching, it is unclear whether the two features are correlated. Dendritic arborization depends primarily on the ORN morphological class. As described in our manuscript, for a given ORN type (T1 for example), we observed noticeable variability in dendritic arborization pattern among T1 ORNs despite their similar soma size.

DiscussionPlease, discuss and cite the relevant literature related your novel findings in more detail. how the size and shape of sensory neurons influence their responses, sensitivity and circuit function of ORNs?

We apologize that our writing was not clear to the reviewer. We have indicated that:

“Indeed, ORNs housed in the same sensillum can inhibit each other by means of direct electrical interaction, termed ephaptic coupling, which can also modulate fruitfly behavior in response to odor mixtures (Su, Menuz, Reisert, and Carlson, 2012; Zhang et al., 2019). Strikingly, in most sensillum types, lateral inhibition is asymmetric between compartmentalized ORNs: the large-spike neuron is not only capable of exerting greater ephaptic influence but is also less susceptible to ephaptic inhibition by its small-spike neighbors. Mechanistically, this functional disparity arises from the size difference between grouped neurons. The large-spike ORN has a larger soma than its small-spike neighbor(s); this feature is translated into a smaller input resistance for the “A” neuron, thus accounting for its dominance in ephaptic interaction (Zhang et al., 2019).”

Other than influencing the strength of ORN ephaptic coupling, how these morphological and morphometric features contribute to sensory function remains an open question.

– New receptors found, e.g. ABx. Give suggestions of how can the be identified (in situ hybridization)?

We are not certain whether in situ hybridization alone will be able to identify the receptors expressed in the neurons of the novel abx(1) or abx(3) sensilla given that their identification likely requires TEM or SBEM examination.

– Some OSNs express more than one receptor. see the work by the Potter lab: Task (2020) Widespread Polymodal Chemosensory Receptor Expression in *Drosophila* Olfactory Neurons 2020.11.07.355651 for discussion discussed in this context.

Although the multiplicity of receptors expressed in individual insect ORNs raises intriguing questions, this information is not directly related to our study. It has been shown that deleting or substituting the tuning OR in an ORN does not change its spike amplitude (Dobritsa et al., Neuron, 2003; Hallem et al., Cell, 2004), which by extension suggests that the receptor does not influence the morphometric feature of an ORN. To explicitly demonstrate this point, we have now provided the information in the revised Introduction “Interestingly, deleting or substituting the tuning receptor for an ORN does not alter its characteristic spike amplitude (Dobritsa, van der Goes van Naters, Warr, Steinbrecht, and Carlson, 2003; Hallem et al., 2004), suggesting that this feature is independent of the receptor identity.”

The ORN in ab4b expresses two receptors: Or56a that respond specifically to goesomin and Or33a that no one could identify its ligand yet.

For clarity, we prefer to use only the tuning receptors to indicate ORN identity without mentioning other co-expressed receptors of unknown function.

– How often did you find extracellular vacuoles within the sensilla lumen, in how many sensilla types they occur?

We thank the reviewer for raising this question. We have now provided detailed information about vacuoles in individual sensilla in the revised Source Data for Table 1.

– Intricate dendritic branching in the outer dendrite ab4A neuron: Figure 8C, and textIn the 2D projection dendritic arbors seems to be fused, that contradicts neuron dichotomy morphology. Is this a simple crossover, or a dendritic tiling shown, here? as described in *Drosophila* larva Grueber (2002).:Tiling of the *Drosophila* epidermis Development, 129(12), see Figure 2B, here. Does the dendritic morphology correlates with arrangement or number of pores in the respective sensillum.

We thank the reviewer for highlighting the novelty of our findings. No, what was described in Figure 8C was not tilting or crossover, as highlighted in Video 7.

Our analysis does not extend to the pore arrangement or numbers. Therefore, we are not able to determine whether there is any correlation between dendritic branching pattern and sensillum cuticular features.

– Empty sensilla containing no neurons, are they described before?

No, they have never been described before, which is why we consider the identification of empty sensilla a novel finding. To clarify the confusion, we have explicitly stated “Such empty sensilla have never been reported before” in the revised text.

Could this phenomen due to a cell turnover, or apoptosis? using P-MARCM Fernández-Hernández (2020). Reported a 3% turnover of ORNs.

We apologize that our writing was not clear to the reviewer. We have already referenced the paper in our original submission, and speculated that empty sensilla may represent a snapshot of ORN turnover. The sentence can be found at the end of the paragraph:

“While the biological significance of empty sensilla remains to be determined, it is possible that they represent a small fraction of antennal olfactory hairs whose neurons have undergone apoptosis in the process of neuronal turnover (Fernández-Hernández, Hu, and Bonaguidi, 2020).”

Concluding remarksDiscuss in more detail your question with the literature what is known. Are empty cells reported before?

No, they have never been described before, which is why we consider the identification of empty sensilla a novel finding. To clarify the confusion, we have explicitly stated “Such empty sensilla have never been reported before” in the revised text.

How your data relate to other studies on the antenna.e.g. Keesey (2019) Inverse location https://www.nature.com/articles/s41467-019-09087-z, for the distribution of sensilla on the antenna.

We wish to clarify that our study focuses on the nanoscale morphological and morphometric features of sensilla and ORNs, instead of the distribution of sensilla on the antenna. Each SBEM volume sampled a specific portion of the antenna covering the majority of APEX2-labeled sensilla, making it difficult to precisely determine its relative antenna location. Therefore, we do not feel comfortable drawing direct comparison to other studies regarding the distribution of sensilla on the antenna.

We did not reference Keesey (2019) because our study concerns only one sensory modality in only one *Drosophila* species. However, we did reference other studies describing the sensillum distribution in an antenna “(de Bruyne, Foster, and Carlson, 2001; Grabe et al., 2016; Shanbhag et al., 1999)”.

Figures– Figure 1A: the distribution on Apex labelled flies (false color?). Did you sample from the anterior / posterior side of the antenna? Please give an orientation, and a relative position of the sampled sensilla, e.g. location of arista, sacculus, and funiculus border (for an nice example, see e.g. Grabe (2016) Figure 1B).– 1B: the columns are show the number of identified sensilla, I suggest, to Indicate in Figure 1 A with stars the labelled versus non-labeled sensilla,

As suggested, we have now indicated APEX2-labeled sensilla in Figure 1A.

– In Figure 1 you gave an overview of datasets used in the study. Please, indicate: how large is your dataset, i.e 123 ORNs, relative to the total number of sensilla (Keesey 2019) reported app. 400 sensilla in total for *Drosophila* m. Also: are your samples randomly chosen across the antenna.Is there a distinct pattern of how are sensilla types are distributed across the antenna for certain sensilla types?

We have indicated the number of sensilla identified in each dataset in Figure 1B. To further address the reviewer’s comment, we have provided the requested information in a new Source Data file for Figure 1.

We wish to clarify that each dataset sampled a specific portion of the antenna. The imaged region was determined by the labeled sensilla, and is therefore not randomly selected.

– Figure 2 the extracellular vacuole (in teal) is shown in dark blue in 2B (3)? I suggest to use another color for the vacuoles for better understanding of the 3D structure.

We thank the reviewer for the excellent suggestion. To better distinguish vacuoles from other structures in our figures, we have changed the vacuole color from teal to aubergine.

– Table 2 (main text): I suggest to provide more details for the identified ORN, to compare with previous datasets, or data repositories (such as DoOR) and to allow for more cross-references found in the literature: – OR co-receptor, – corresponding glomerulus in the antennal lobe, – functional response according to DoORIn the DoOR database (http://neuro.uni-konstanz.de/DoOR/default.html ), for example, the ab4 sensillum hosing Or7A receptor (ab4A neuron), and the Or56a co-expressed with Or33Aa. The latter is not listed in your table.

As suggested, we have included ORN glomerulus projection as an additional identifier. The revised legend now includes: “ORN identity is indicated by the sensillum type, relative spike amplitude (A, B, C or D), odor-tuning receptor, and glomerular projection.”

For clarity, and in consideration of the large information load, we focus on our own morphometric data in Table 2. For the same reasons, we also focus on the tuning receptors as a key ORN identifier without mentioning other co-expressed receptors of unknown function.

Furthermore, we wish to clarify that the odor response profile of an ORN is predominantly determined by the tuning receptor expressed in the neuron (Hallem et al., Cell, 2004), which is independent of whether the neuron is a large- or small-spike ORN. By extension, the odor response profile is independent of the neuron’s morphometric features. It is therefore of limited usefulness to search for any correlation between ORNs’ morphological features and odor response properties.

However, we have incorporated the reviewer’s suggestion by revising the text to include key ligands for ORNs that respond to ethologically salient odors. We have also included the following sentence in the revised Table 2 legend “The odor response profiles for many of the characterized ORNs can be found in the DoOR database (http://neuro.uni-konstanz.de/DoOR/default.html)” such that readers who are curious about the functional data can easily find the information.

SupplementarySu_28-04-2021-RA-eLife-69896_Source_Data_File excel TableIn general: provide an explanation and an list of all abbreviation used in these excel files. There are partly hard to understand.– ORN Morphometric: I do not understand the terminology of column 2

We are sorry for the confusion. The second column refers to the sensillum identity. In the revision, we have provided a clearer definition for all abbreviations and terms in the Source Data for Table 2.

– T3 model: why you present data three soma sizes: A,B,C, but listing only two ORN types in the first column ? same for AC1 model

We are sorry that we do not understand what the reviewer referred to. The first column indicates the dataset identity, and the second column refers to the sensillum identity.

– It took me a while to figure out that the first column 'just' indicates the dataset (genetically labelled with Apex) without any other reference to the following columns identity of ORN , and labelled with A,B, or C.I suggest to present, here, and in the tables where you report 'branching' and 'mitochondria' the ORN type in brackets, as you do in the Summary Table.

We are sorry for the confusion. In the revision, we have provided a clearer definition for all abbreviations and terms in the Source Data for Table 2.

Videos: Please, prepare an extra sheet, word-file to provide a detailed description for each video

We thank the reviewer for the suggestion. We have now provided detailed information for each 3D model shown in the videos in the Source Data for Table 2 and Source Data for Figure 1. The video legend now includes “Detailed information regarding individual 3D models shown in the videos can be found in the “Source Data for Table 2” file.”